# Does Thinking More *Always* Help?
# Mirage of Test-Time Scaling in Reasoning Models

**Soumya Suvra Ghosal**[1*]**, Souradip Chakraborty**[1*]**, Avinash Reddy**[5]**, Yifu Li**[3]**,**
**Mengdi Wang**[3]**, Dinesh Manocha**[1]**, Furong Huang**[1,3]**,**
**Mohammad Ghavamzadeh**[4]**, Amrit Singh Bedi**[5]
[1]University of Maryland, [2]Princeton University,
[3]Capital One, [4]Amazon AGI, [5]University of Central Florida

## Abstract

Recent trends in test-time scaling for reasoning models (e.g., OpenAI o1, DeepSeek R1) have led to a popular belief that extending thinking traces using prompts like "Wait" or "Let me rethink" can improve performance. This raises a natural question: *Does thinking more at test-time truly lead to better reasoning?* To answer this question, we perform a detailed empirical study across models and benchmarks, which reveals a consistent pattern of initial performance improvements from additional thinking followed by a decline, due to '*overthinking*'. To understand this non-monotonic trend, we consider a simple probabilistic model, which reveals that additional thinking increases output variance, creating an illusion of improved reasoning while ultimately undermining precision. Thus, observed gains from "more thinking" are not true indicators of improved reasoning, causing a *mirage effect*, but artifacts stemming from the connection between model uncertainty and evaluation metric. This suggests that test-time scaling through extended thinking is not an effective way to utilize the inference thinking budget. Recognizing these limitations, we introduce an alternative test-time scaling approach, *parallel thinking*, inspired by Best-of-N sampling. Our method generates multiple independent reasoning paths within the same inference budget and selects the most consistent response via majority vote, achieving up to 20% higher accuracy compared to extended thinking. This provides a simple yet effective mechanism for test-time scaling of reasoning models.

## 1 Test-time Scaling Capability in LLMs for Reasoning

Recent improvements in the performance of large language models (LLMs) on reasoning tasks have been primarily driven by enhancements in their reasoning capabilities through reinforcement learning-based training (Guo et al., 2025; Shao et al., 2024). With increasingly capable reasoning models, a key question that arises is how to effectively scale their performance at test time. Recent literature (Muennighoff et al., 2025) has shown that extending the thinking (or reasoning) process at test-time through explicit prompts such as *"Wait"*, *"Let me rethink"*, or *"Think more"* significantly improves performance. This ap-

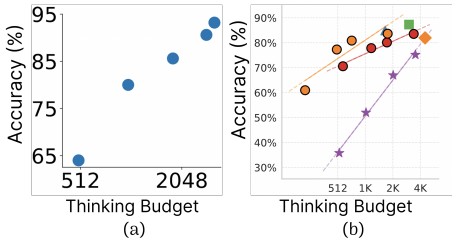

Figure 1: (a) Scaling behavior observed in Muennighoff et al. (2025). (b) Trends in Aggarwal & Welleck (2025) (on MATH-500 (Hendrycks et al., 2021b)).

---

[*]Equal Contribution.

39th Conference on Neural Information Processing Systems (NeurIPS 2025).

proach, requiring no additional supervision, has led
to the prevailing belief that more thinking at test-time inherently improves accuracy (see Figure 1).

**Incomplete picture of test-time scaling**. While appealing, the above narrative in prior works presents an incomplete picture. In contrast to prior claims, our empirical investigation uncovers a nuanced phenomenon: extending thinking at test-time initially boosts model accuracy, but performance degrades subsequently with prolonged thinking (cf. Section 2.3). This non-monotonic behavior (a clear pattern consistent across various tasks and datasets (cf. Figure 2) reveals the presence of a *critical point* in the length of the thinking trace beyond which performance declines, which we call 'overthinking', largely unrecognized by existing research. These observations raise a fundamental question: *Why does additional thinking beyond a certain point degrade the model performance?*

**Understanding overthinking: a variance-based explanation.** To answer the above question, we take a step back and analyze a simple one-dimensional probabilistic framework (cf. Section 3), examining how changes in the variance of the sampling distribution affect the expected value of a target reward. Interestingly, we observe that as the variance increases from low to high, the expected reward exhibits a similar non-monotonic pattern: initially increasing, then decreasing (cf. Figure 4). Inspired by this insight, we empirically assess the variance of reasoning-model outputs under extended thinking by measuring the entropy of their output distributions. Our results clearly demonstrate that extended thinking significantly increases the variance of response distribution (cf. Figure 5). This explains why average accuracy first improves and then deteriorates, revealing that the apparent gains from extended thinking reflect an illusion rather than genuine improvements in reasoning capability.

**Overthinking is inefficient for test-time scaling under a fixed budget.** These insights reveal a deeper inefficiency: extending a single reasoning trace is not an optimal use of the test-time compute budget. Because performance does not improve monotonically with more tokens, there is no reliable stopping criterion, making this strategy brittle in practice.

**A fix: parallel thinking as a principled alternative.** To overcome these limitations, we propose *parallel thinking*, a test-time scaling strategy inspired by Best-of-$N$ sampling (Beirami et al., 2024; Amini et al., 2024; Nakano et al., 2021; Stiennon et al., 2020; Gui et al., 2024; Jinnai et al., 2024). Instead of continuing one thinking trace, we allocate the same token budget across multiple independent thinking paths and select the final answer via majority voting. This approach avoids entropy overgrowth, mitigates the overthinking trap, and achieves significantly better performance. For example, under a 16K token budget, parallel thinking yields up to 22% higher accuracy compared to sequential scaling (Figure 7). We summarize our contributions as follows.

**(i) Empirical diagnosis of overthinking:** We investigate test-time scaling by encouraging extended thinking in state-of-the-art reasoning models with prompts as "Wait", "Think more" etc. It reveals a consistent non-monotonic trend in performance across multiple tasks and datasets (cf. Section 2).

**(ii) Illusion of test-time scaling: alternative explanation**: We provide an interesting explanation for the non-monotonic trend of test-time scaling in reasoning models through a simple probabilistic framework. Our analysis clarifies why extending reasoning initially improves performance but eventually leads to degradation, highlighting variance as the key driver of the observed non-monotonic behavior (cf. Section 3).

**(iii) Variance-driven explanation of performance degradation in reasoning models:** By analyzing the entropy of the response distribution generated by the reasoning models, supported by our insights, we show that extended thinking increases the variance of the model's output distribution. While this variance increase initially aligns with improved performance, it eventually disrupts reward alignment, explaining the degradation observed beyond a certain point (cf. Section 3.1).

**(iv) Effective budget-control via parallel thinking:** We propose an alternative test-time scaling strategy, *parallel thinking*, inspired by Best-of-$N$ sampling. By simultaneously generating multiple independent reasoning paths, this approach circumvents the pitfalls of sequential overthinking and yields higher performance, demonstrating genuine self-improvement capabilities (cf. Section 4). This approach outperforms overthinking across all benchmarks and provides a reliable mechanism for inference-time scaling (Figure 7).

## 2 Analysis of Test-Time Scaling in SoTA Reasoning LLMs

**Mathematical formulation of thinking process**: We begin with a mathematical representation of the thinking process generated by reasoning models, specified as $x \rightarrow z \rightarrow y$,

where $x$ is the input prompt, $z \sim \pi(\cdot|x)$ is the intermediate thinking trace, $y \sim \pi(\cdot|x, z)$ is the final answer, and $\pi$ denotes the language model. The RL objective for training reasoning models is:

$$\max_\theta \mathbb{E}_{x, z \sim \pi_\theta(\cdot|x), y \sim \pi_\theta(\cdot|x,z)}[R(x, y)], \tag{1}$$

where $\pi_\theta$ is the parameterized model and $R(x, y)$ represents the true reward function (for example, an indicator to check if $y$ is correct or not), which is obtained once the policy generates the final response $y$. In practice (Guo et al., 2025), an additional format reward term, $R_{\text{format}}(x, y)$, is also often incorporated to encourage the model to generate outputs in a structured format: $x \rightarrow z \rightarrow y$. That is to make sure that the model first produces a "thinking" trace $z$, followed by a special end-of-thinking token (`</think>`), and then the final answer $y$. This process of first generating a thinking trace and then answering $x \rightarrow z \rightarrow y$ is referred to as "standard thinking" in this work.

**Test time scaling.** The test-time scaling extends the thinking process as

$$x \rightarrow z_1 \rightarrow z_2 \rightarrow \cdots \rightarrow z_k \rightarrow y, \tag{2}$$

where, given the prompt $x$, the first thinking step is generated as $z_1 \sim \pi_\theta(\cdot|x)$. Then, instead of directly producing the final answer, the model is instructed to continue thinking using special tokens such as *"Wait"* or *"Think more"*, denoted as p with the next thinking step sampled as $z_2 \sim \pi_\theta(\cdot|x, z_1, p)$. The process is continued iteratively for $k$ steps, and the model generates the final response as $y \sim \pi_\theta(\cdot|x, z_1, p, z_2, p, \ldots, z_k)$. For simplicity of representation, we just keep the thinking traces in the condition as $y \sim \pi_\theta(\cdot|x, z_1, z_2, \ldots, z_k)$, omitting the special instruction tokens $p$.

As discussed in the introduction, to gain a complete understanding of test-time scaling in reasoning language models, we begin with a systematic analysis of state-of-the-art reasoning models (Guo et al., 2025). This analysis requires two core components: the choice of reasoning models and the strategies used to control the test-time compute budget. We first introduce the budget control techniques we evaluate, then describe the reasoning models under study, and finally highlight key takeaways and novel insights that emerge from this investigation.

### 2.1 Test-Time Budget Control (TTBC)

To systematically probe how test-time budgets modulate model behavior, we apply two different budget control approaches on the model's *thinking* tokens (Muennighoff et al., 2025) detailed as follows.

- **TTBC 1: Wait & Think more.** In this approach, we do not impose any explicit budget constraint on the number of thinking tokens, apart from the model's inherent maximum token limit ($32K$). Specifically for this approach, whenever the model attempts to generate the end-of-thinking delimiter (`</think>`), we suppress it, append the token *"Wait"* to the thinking trace, and feed the modified trace back to the model. This intervention is applied iteratively, encouraging the model to extend its thinking. In this setup, there is no explicit budget for the thinking tokens, except the max token limit of the model, and we only modify the number of times *"Wait"* is appended to the thinking trace.

- **TTBC 2: Exact thinking tokens.** We enforce an exact thinking token budget of $t_{\text{exact}}$ for each reasoning trajectory. Specifically, we iteratively append the token *"Wait"* to the reasoning trace until the cumulative count of thinking tokens reaches exactly $t_{\text{exact}}$. Once this threshold is reached, we terminate the thinking stage by allowing the end-of-thinking delimiter (`</think>`) to pass, signaling the model to generate the final response. This approach ensures that every reasoning trajectory is precisely constrained to $t_{\text{exact}}$ tokens. For this setup, we vary $t_{\text{exact}}$ across $[256, 512, 1024, 2048, 4096, 8192, 16384]$.

### 2.2 Experimental Setup

**Datasets.** Following prior literature (Arora & Zanette, 2025; Aggarwal & Welleck, 2025), we base our analysis on three widely-adopted mathematical reasoning datasets: (a) GSM-8K (Cobbe et al.,

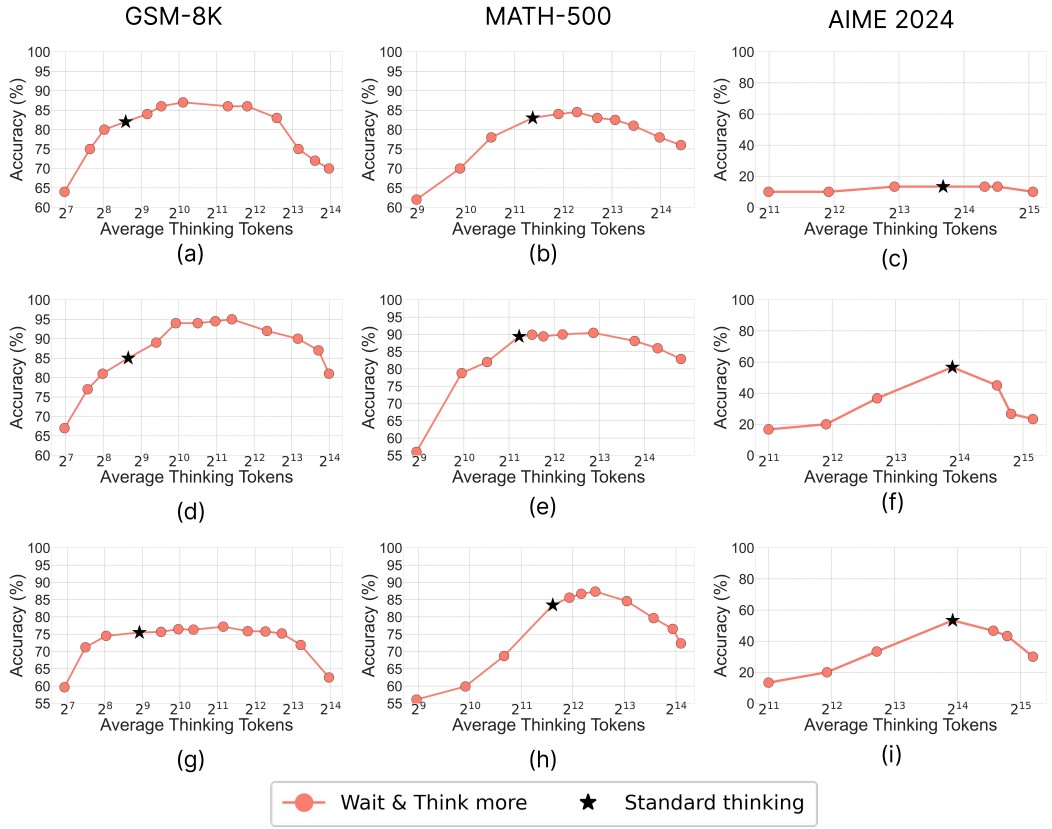

Figure 2: **Results for *Wait & Think more*.** To investigate the variation in accuracy with increasing average thinking tokens generated during inference, we plot accuracy as a function of thinking tokens across three datasets: GSM-8K (Cobbe et al., 2021), MATH-500 (Lightman et al., 2023), and AIME (invitational mathematics examination, 2024). The average has been taken over all prompts in the test dataset. We report results for three SoTA open-source reasoning models: DeepSeek-R1-Distill-Qwen-1.5B (1st row), DeepSeek-R1-Distill-Qwen-7B (2nd row), and DeepSeek-R1-Distill-Llama-8B model (3rd row) (Guo et al., 2025). During test-time, following Muennighoff et al. (2025), we induce the model to generate more thinking tokens by suppressing the end-of-thinking token delimiter (`</think>`) and instead appending *"Wait"* to the model's current thinking trace to encourage additional thinking. The baseline with no budget control (standard thinking) is marked by ⋆. Successive **salmon circles** to the right of the star correspond to one, two, … additional *"Wait"* insertions to the model's thinking trace. We note that to obtain the circles on the left of the ⋆, we enforce budgeting via controlling the maximum number of thinking tokens. From our experiments, we note that accuracy initially rises as the model is prompted to think longer, but then gradually falls off for all settings.

2021), a dataset consisting of grade-school math problems that typically require multi-step reasoning through basic arithmetic operations to arrive at the correct answer. For this study, we evaluate on the 1,320 samples from the GSM-8K test set. (b) MATH-500 (Hendrycks et al., 2021b) is a benchmark of challenging math problems of varying difficulty. For our study, we selected the same 500 samples selected by OpenAI in (Lightman et al., 2023), and (c) AIME 2024 invitational mathematics examination (2024), a set of 30 competition-level problems from the American Invitational Mathematics Examination 2024, designed to assess advanced problem-solving skills across topics like arithmetic, algebra, geometry, number theory, probability, and other secondary school mathematics.

**Reasoning language models.** To ensure reproducibility, we leverage open-source reasoning models for our analysis. Specifically, we employ three models of varying sizes and architectures, each distilled from the recently released DeepSeek-R1 model (Guo et al., 2025) by Guo et al. (2025): DeepSeek-R1-Distill-Qwen-1.5B, DeepSeek-R1-Distill-Qwen-7B, and the Llama-based variant DeepSeek-R1-Distill-Llama-8B.

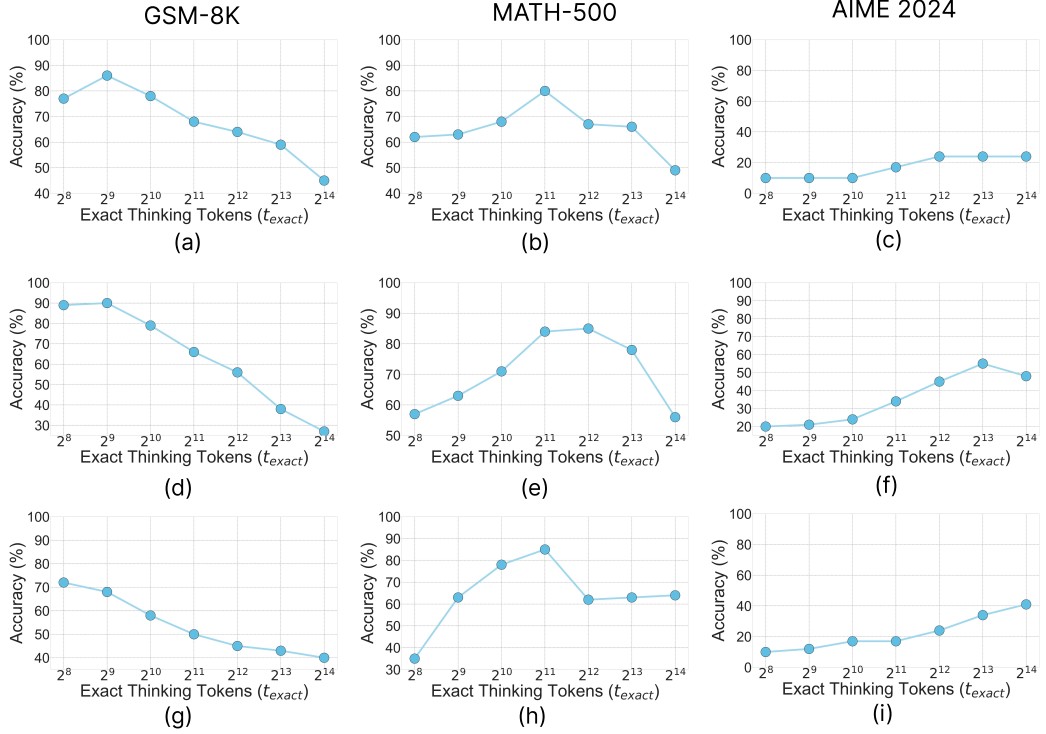

Figure 3: **Results for *Exact thinking tokens* setup.** We visualize accuracy as a function of the thinking token budget for three benchmark datasets: GSM-8K (Cobbe et al., 2021), MATH-500 (Lightman et al., 2023), and AIME (invitational mathematics examination, 2024). We report results for two open-source reasoning models: DeepSeek-R1-Distill-Qwen-1.5B (1st row), DeepSeek-R1-Distill-Qwen-7B (2nd row), and DeepSeek-R1-Distill-Llama-8B (3rd row) (Guo et al., 2025). For each prompt, we iteratively append the token *"Wait"* to the reasoning trace until the cumulative count of thinking tokens reaches exactly $t_{\text{exact}}$. We consistently observe a decline in accuracy with an increase in the thinking token budget.

**Evaluation Criteria.** To evaluate reasoning performance, we report the accuracy of each model on the test set of each dataset. Specifically, for each prompt $x \in \mathcal{D}^{\text{test}}$, the model first generates a thinking trace $z$, followed by the final answer $y$. The accuracy is then defined as: $\mathbb{E}_{x \sim \mathcal{D}^{\text{test}}, z \sim \pi_\theta(\cdot|x), y \sim \pi_\theta(\cdot|x,z)} \left[ \mathbb{I}\{y = y^*\} \right]$, where $y^*$ is the correct answer.

## 2.3 Key Observations: Failure of Overthinking

In Figure 2, we illustrate the trade-off between average thinking tokens and average accuracy under the *Wait & Think More* TTBC. We observe an initial increase (similar to (Muennighoff et al., 2025; Aggarwal & Welleck, 2025)) in accuracy as the average thinking budget increases. For example, in Figure 2(a), accuracy increases from $82.2\%$ to $87.3\%$ as the average number of thinking tokens increases from 385 to 1100. However, this trend does not continue indefinitely. Beyond a critical point, further increasing the thinking budget results in a steady decline in accuracy. Specifically, pushing the average thinking token count from 1100 to 15980 reduces accuracy from $87.3\%$ to $70.3\%$.

This observed non-monotonic trend challenges the prevailing assumption that *"more thinking is always better."* Instead, it reveals a more nuanced insight: test-time reasoning exhibits a *critical spot*, a point beyond which additional thinking transitions from helpful to not being helpful anymore- a phenomenon we call 'overthinking'. Prior work has overlooked this degradation phase, presenting an incomplete view of the true test-time scaling landscape. To further understand and explain this phenomenon, we extend our analysis in Figure 3 using the Exact Thinking tokens setup (TTBC 2). These findings consistently reaffirm the same trend, underscoring the importance of reconsidering current test-time reasoning strategies and moving beyond the simplistic belief that more computation inherently leads to better reasoning outcomes.

In Appendix E.1, we extend this analysis to the Minimum Thinking tokens TTBC, and observe a similar non-monotonic relationship between accuracy and the number of thinking tokens.

## 3  Mirage of Test-Time Scaling: An Alternative Explanation

In this section, we propose an alternative explanation for the observed non-monotonic behavior in test-time scaling of reasoning models using a simple probabilistic framework. Although state-of-the-art reasoning models initially show accuracy improvements with increased test-time thinking, they consistently exhibit performance deterioration after exceeding a specific point. This observation prompts a fundamental question: Is the performance gain from additional thinking genuinely indicative of enhanced reasoning capability, or merely a mirage? We illustrate the mirage effect of test-time scaling via a simple probabilistic illustration below.

**Setup.** To illustrate this mirage effect, we consider a simplified scenario. Assume the reward is fixed and unimodal, and the model samples answers from the policy distribution $y \sim \pi(\cdot|x)$. Sepcifically, we assume $\pi(y|x) = \mathcal{N}(\mu_\pi, \sigma_\pi^2)$ (a univariate Gaussian distribution) and reward function follows a bell-curve as $r(x, y) = \mathcal{N}(\mu_r, \sigma_r^2)$. Hence, we can write

$$\mathbb{E}_{y \sim \pi(\cdot|x)}[r(x, y)] = \int r(x, y) \cdot \pi(y|x) \cdot dy. \tag{3}$$

From the assumptions on the policy $\pi$ and reward $r$, we can obtain

$$\mathbb{E}_{y \sim \pi(\cdot|x)}[r(x, y)] = \underbrace{\frac{1}{\sqrt{2\pi(\sigma_r^2 + \sigma_\pi^2)}}}_{\text{Density Spread}} \cdot \underbrace{\exp\left(-\frac{(\mu_r - \mu_\pi)^2}{2(\sigma_r^2 + \sigma_\pi^2)}\right)}_{\text{Mean Alignment Penalty}}. \tag{4}$$

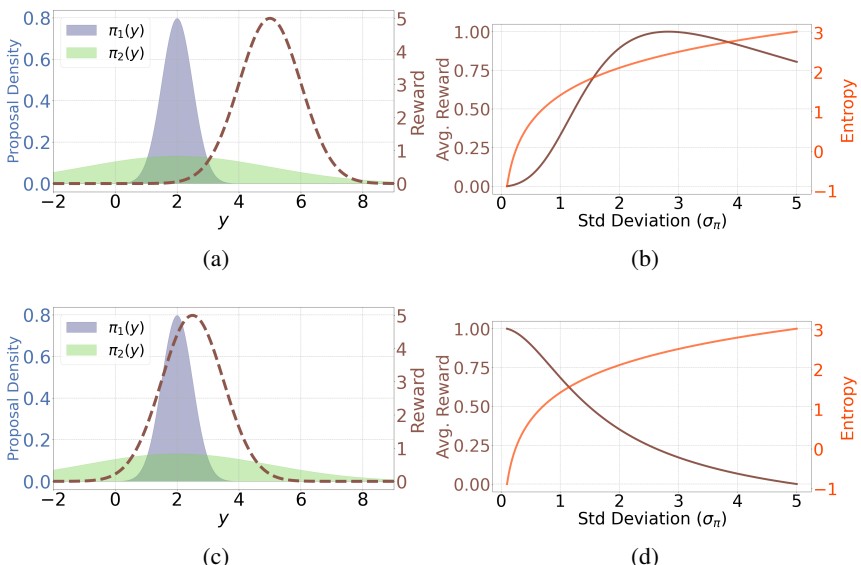

(a)   (b)

(c)   (d)

Figure 4: **A simple illustration of the mirage in test-time scaling. Scenario 1:** (top row) When the mean of the proposal policy and reward function is far away (cf. (a)), as the variance of the proposal increases, overlap with the reward improves initially but degrades past a sweet spot, shown by the peak in reward (brown line in (b)). **Scenario 2:** When the mean of the proposal policy and reward function is well aligned (cf. (c)), increasing variance results in diminished reward with continued entropy growth (cf. (d)).

**Effect of increase in variance $\sigma_\pi^2$.** As shown in Figures 4, we observe that initially increasing $\sigma_\pi^2$ helps: it broadens $\pi(y|x)$, increasing its overlap with the reward distribution $r(x, y)$. As a result, the expected reward increases, leading to improved performance. However, this trend does not continue indefinitely, as performance begins to degrade beyond a certain point with more increase in variance. When the variance is too small, the model remains stuck near the proposal mean, resulting in poor

reward due to limited exploration. Conversely, when the variance is too large, the model samples indiscriminately across the space, again leading to poor reward. Thus there exists a *critical point* in the variance of the proposal distribution.

**Why does the critical point exist?** It arises because of two competing forces: *Coverage effect:* For small $\sigma_\pi^2$, increasing variance improves average reward by covering more of the reward peak centered at $\mu_r$. *Dilution effect:* Beyond a point, increasing variance overspreads the distribution, placing mass on regions far from $\mu_r$, leading to diminished expected reward. The trade-off is evident in equation 4, where initially, the exponential term dominates: increasing $\sigma_\pi^2$ helps reduce the exponent (denominator increases), improving expected reward. Eventually, the prefactor $1/\sqrt{2\pi(\sigma_r^2 + \sigma_\pi^2)}$ shrinks faster than the exponent gains, reducing the overall value.

**Connection with the test-time scaling in reasoning models.** Our simple illustration reveals a powerful insight: increasing the variance of the policy distribution can initially boost expected reward, not because the policy has improved, but due to greater overlap with the reward. Crucially, this improvement is a mirage, driven by randomness rather than genuine policy refinement. We hypothesize that a similar effect underlies the observed gains in test-time scaling of reasoning models observed in Figure 2. Specifically, increasing thinking via longer thinking traces with prompts like "Wait" acts as a knob to affect the variance of the model's output distribution.

In the next subsection, we draw this analogy directly: each additional reasoning/thinking step increases the entropy of the policy, leading to broader sampling and a rise in accuracy, up to a point. Beyond that, the distribution becomes too diffuse, and performance deteriorates.

### 3.1 Role of Variance in test-time scaling of reasoning models

In this section, we empirically investigate the effect of extended thinking on model uncertainty and its role in the mirage phenomenon (Section 3). Specifically, our goal is to analyze how increasing the length of thinking tokens influences the entropy of the policy distribution.

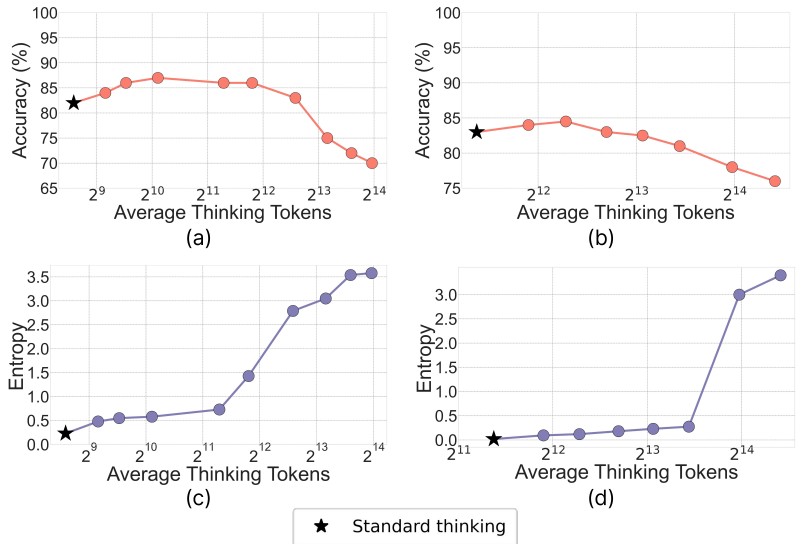

Figure 5: **Role of variance in test-time scaling.** We analyze how extended thinking influences policy uncertainty by visualizing entropy as a function of additional thinking tokens on two benchmarks: GSM-8K (Cobbe et al., 2021) (Figures (a) and (c)) and MATH-500 (Hendrycks et al., 2021b) (Figures (b) and (d)). We observe that with additional thinking, there is an initial rise in entropy, leading to improved performance. However, beyond a critical point, extended thinking results in a steep rise in entropy, adversely affecting performance. For this analysis, we used the "Wait & Think more" setup with the DeepSeek-R1-Distill-Qwen-1.5B model.

For this analysis, we evaluate the DeepSeek-R1-Distill-Qwen-1.5B model (Guo et al., 2025) on two mathematical reasoning benchmarks: GSM-8K (Cobbe et al., 2021) and MATH-500 (Hendrycks et al., 2021b). We adopt the *Wait & Think more* setup, where we suppress the generation of the end-

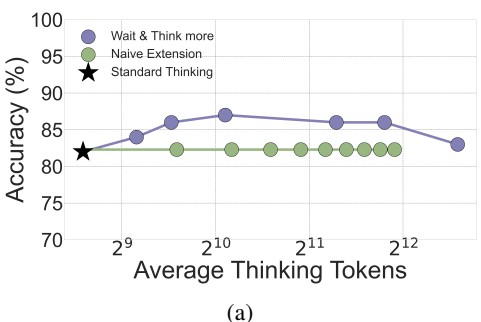 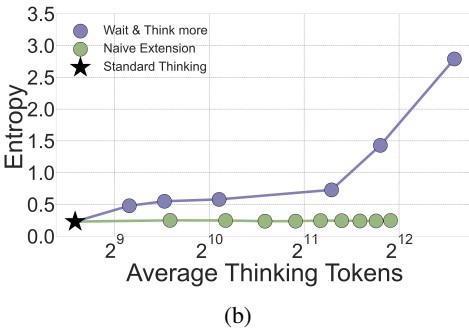

(a)                              (b)

Figure 6: **Understanding the impact of context length on generation.** To investigate whether increased variance originates from longer input sequences or from overthinking, we performed an experiment on GSM-8K using DeepSeek-R1-Distill-Qwen-1.5B. Rather than sampling diverse reasoning traces, we repeatedly appended the same reasoning step, $z_1 \sim \pi(\cdot \mid x)$, to artificially extend the input before generating the final answer $y \sim \pi(\cdot \mid x, z_1, z_1, \ldots, z_1)$. The entropy of the resulting response distribution remained nearly unchanged, demonstrating that simply increasing context length does not induce greater uncertainty.

of-thinking delimiter and append *"Wait"* to encourage the model for additional thinking. To measure the entropy of the policy, after appending each *"Wait"*, we sample multiple answers $y \sim \pi(\cdot|z_{1:i}, x)$, where $z_{1:i} = [z_1, \cdots, z_{i-1}, z_i]$ denotes the cumulative sequence of thinking tokens generated up to the $i$th-*"Wait"*, $z_1$ denotes the initial thinking trace generated with standard thinking, and $x$ is the given prompt. We quantify policy uncertainty using the entropy of the distribution over final answers: $\mathbb{E}_{y \sim \pi(\cdot|z_{1:i}, x)}[-\log \pi(y \mid z_{1:i}, x)]$.

We report the results for this analysis averaged over all prompts in the test dataset in Figure 5, where baseline entropy for standard thinking (without any additional *"Wait"*) is marked by a ⋆ (**black star**). We observe: (1) For both benchmarks, the entropy of the policy distribution increases as the model is prompted to think longer. For example, in Figure 5 (c) for GSM-8K, extending the average thinking tokens from 385 (standard thinking) to 6136 resulted in a 12x increase in entropy, from 0.23 to 2.79. (2) As discussed in Section 3, this initial rise in entropy leads to an initial improvement in average accuracy. For instance, for MATH-500 in Figure 5 (d), the initial increase in entropy from 0.02 to 0.12 (with thinking extended from 2657 to 4983 tokens) corresponds to a 1.5% accuracy gain, as shown in Figure 5 (b). However, with further thinking (from 4983 to 16109 tokens), there is a steep rise in the entropy of the policy distribution from 0.12 to 3.0, indicating an increase in the policy's uncertainty. This rise in entropy also correlates with a decline in average accuracy from 83.2% with standard thinking to 78.3% (see Figure 5 (b)). We extend this analysis to other models in Appendix E.2, where we observe a similar trend of increasing entropy with additional thinking tokens.

**Key takeaway and budget control problem:** Our investigation reveals that the initial accuracy gains from test-time scaling up to some average thinking tokens are largely a mirage, arising from randomness rather than the model's refinement. This observation further implies that the *"Wait & Think More"* approach is not an optimal way to utilize the test-time thinking budget, which leads to a fundamental question: *Given a fixed thinking budget, how to optimally allocate resources for test-time scaling in reasoning models?* In the next section, we provide a solution to effective budget control.

To gain deeper insight into the source of increased variance during overthinking, we conducted an additional experiment on GSM-8K using DeepSeek-R1-Distill-Qwen-1.5B. Specifically, instead of extending the chain-of-thought through repeated sampling of diverse reasoning traces, we artificially lengthened the reasoning sequence by repeatedly appending the same reasoning step, $z_1 \sim \pi(\cdot \mid x)$, multiple times before generating the final answer $y \sim \pi(\cdot \mid x, z_1, z_1, \ldots, z_1)$. We then measured the entropy of the resulting response distribution, as reported in Figure 6 (b). The results show that entropy does not increase simply by lengthening the input with repeated tokens. This finding suggests that it is not just the context length but the overthinking that affects the performance.

## 4   Effective Test-Time Budget Utilization via Parallel Thinking

In this section, we propose a simple approach for efficiently utilizing the given thinking token budget (say $B$ tokens) by generating multiple parallel streams of thoughts. To be specific, instead of spending

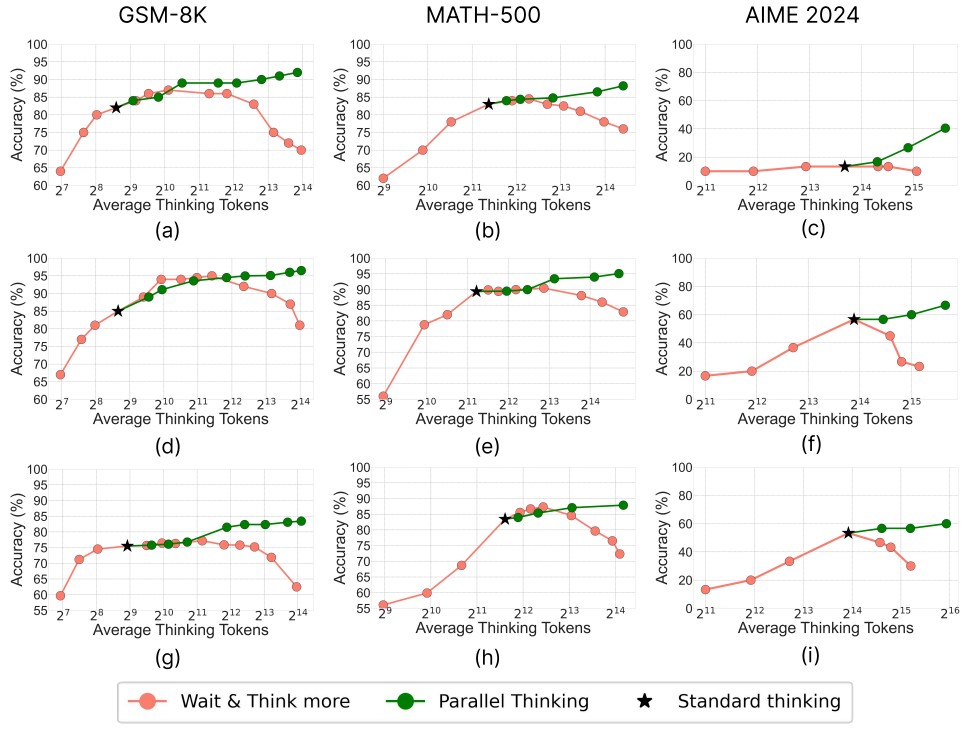

Figure 7: **Comparing test-time scaling strategies.** We compare test-time scaling using *"Wait & Think more"* and parallel thinking across three benchmark datasets (GSM-8K (Cobbe et al., 2021), MATH-500 (Hendrycks et al., 2021b), and AIME 2024 (invitational mathematics examination, 2024)) and three state-of-the-art reasoning models: DeepSeek-R1-Distill-Qwen-1.5B (first row), DeepSeek-R1-Distill-Qwen-7B (second row), and DeepSeek-R1-Distill-Llama-8B (third row). Each plot shows average accuracy (y-axis) against average thinking tokens (x-axis) as the thinking budget is progressively increased, with averages taken across all test set prompts. Baseline performance without budget control (standard thinking) is marked by a **black star**. Additional thinking using *"Wait & Think More"* is shown as **salmon circles**, while parallel scaling is represented by **green circles**. Unlike sequential scaling, which can degrade performance with overthinking, parallel scaling consistently maintains or improves accuracy as the budget increases.

the entire budget $B$ on a single extended chain of thought with $B$ tokens, we use the model $\pi(\cdot|x)$ to generate $N$ parallel thinking traces $z^{(i)} \sim \pi(\cdot \mid x), \quad i = 1, \ldots, N$ such that $\sum_{i=1}^{N} |z^{(i)}| \leq B$, where $B$ is the thinking budget. Next, for each reasoning trace $z^{(i)}$, we sample a single output $y^{(i)} \sim \pi(\cdot \mid z^{(i)}, x)$. From the resulting candidate set $\mathcal{Y} = \{y^{(1)}, y^{(2)}, \ldots, y^{(N)}\}$, we select the final output as: $y^{\text{best}} = \arg\max_{y \in \mathcal{Y}} R_{\text{self}}(x, y)$, where we define $R_{\text{self}}(x, y) := \sum_{i=1}^{N} \mathbb{I}[y^{(i)} = y]$ and $\mathbb{I}$ is the indicator function. The rationale for employing this self-reward mechanism is to favor high-likelihood responses. While directly generating the optimal (i.e., highest probability) response from an autoregressive language model is known to be NP-hard in the worst-case scenario Huang et al. (2024a), evaluating the model's likelihood for a given response $y^{(i)}$ (conditioned on $x$ and $z^{(i)}$) is straightforward. Hence, we leverage this tractable method for identifying and selecting high-reward responses from the set of generated candidates.

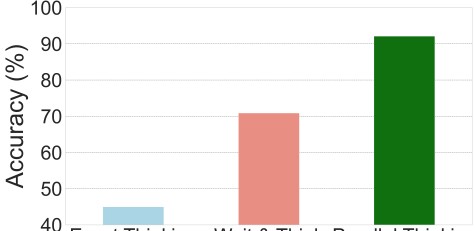

Figure 8: **Effective utilization of test-time budget.** Given a fixed inference budget of $16K$ tokens, parallel thinking results in around $22\%$ and $47\%$ higher accuracy compared to *Wait & Think More* and *Exact Thinking TTBC*, respectively.

**Empirical evaluations.** In Figure 7, we compare sequential thinking using "Wait & Think more" and parallel thinking across three benchmark datasets: GSM-8K (Cobbe et al., 2021), MATH-500 (Hendrycks et al., 2021b), and AIME 2024 (invitational mathematics examination, 2024), and three state-of-the-art reasoning models. We note that, unlike sequential scaling, where extended thinking can degrade model performance, parallel thinking consistently maintains or enhances accuracy with increasing thinking budget. For example, in Figure 7(a), when evaluating the DeepSeek-R1-Distill-Qwen-1.5B model on GSM-8K, increasing the thinking budget from 385 tokens (standard thinking) to 16000 tokens using the *"Wait & Think More"* approach results in a 11.8% performance drop. In contrast, utilizing the same thinking token budget through parallel thinking boosts accuracy by 10.1% compared to standard thinking.

## 5   Related Works

**Large reasoning models.** OpenAI's o1 (OpenAIo1, 2024) series introduced the first large-scale language model to integrate RL-driven training with the concept of test-time thinking, unlocking the reasoning potential of LLMs, and establishing the domain of *large reasoning models* (Xu et al., 2025). Following the release of o1, numerous efforts have sought to replicate its reasoning capabilities, including Open-R (Wang et al., 2024b), Llama-berry (Zhang et al., 2024a), Journey-R (Qin et al., 2024), and Drt-o1 (Wang et al., 2024a). These approaches typically leverage Monte-Carlo Tree Search (MCTS) (Gao et al., 2024; Qin et al., 2024; Zhao et al., 2024; Zhang et al., 2024b,a) or distillation from reasoning traces generated by o1-like models (Huang et al., 2024b). One of the first open-source models to match o1-level reasoning performance was DeepSeek-R1 (Guo et al., 2025), which employed a sparse-reward based RL training framework (Shao et al., 2024) on the DeepSeek-V3-Base model (DeepSeek-AI, 2024) to enhance its reasoning capabilities. Following DeepSeek-R1, recently several open-source efforts have continued employing RL training (Shao et al., 2024) to improve reasoning performance (Team, 2025; Arora & Zanette, 2025; Aggarwal & Welleck, 2025; Liu et al., 2025; Shen et al., 2025; Yu et al., 2025; Huang et al., 2025b; Jin et al., 2025; Team et al., 2025; xAI, 2025; Google DeepMind, 2024). In this study, we specifically focus on analyzing the test-time scaling behavior in reasoning models.

**Test-time scaling in reasoning language models.** Recent work by Muennighoff et al. (2025) introduced the concept of budget forcing to replicate the test-time scaling behavior observed in o1 models (OpenAIo1, 2024). Another recent approach, L1 (Aggarwal & Welleck, 2025), proposed length-controlled policy optimization, providing precise control over the length of the reasoning trace during generation. Yang et al. (2025) introduced a thinking-optimal scaling strategy, training models to adapt dynamically to different levels of reasoning effort depending on the test-time compute budget. Recently, a lot of studies have also focused on fine-tuning models to think efficiently according to task complexity (Arora & Zanette, 2025; Fang et al., 2025; Zhang et al., 2025b; Jiang et al., 2025; Liang et al., 2025; Zhang et al., 2025a; Huang et al., 2025a).

## 6   Conclusions and Limitations

In this paper, we present a systematic study of test-time scaling in reasoning models, analyzing how prompting models to *"Wait & think"* affects performance. Our key finding is that while extended thinking initially improves accuracy, continued reasoning beyond a threshold degrades performance—a phenomenon we term *overthinking*. We show that these early gains are often illusory, stemming from increased response variance rather than genuine reasoning improvements. To address this, we propose a simple and effective alternative: *parallel thinking*, which distributes the inference budget across multiple reasoning paths and selects the final answer via majority vote, yielding consistent gains.

**Limitations.** Our study is limited to mid-sized models due to resource constraints; evaluating the behavior of larger models (e.g., 32B and 70B) remains an exciting avenue for future work. Moreover, while our insights are supported by empirical evidence and a simple probabilistic model, developing a formal theoretical understanding of how variance and entropy affect reasoning performance is an important direction for deeper investigation.

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

# A Acknowledgments

Chakraborty and Huang are supported by DARPA Transfer from Imprecise and Abstract Models to Autonomous Technologies (TIAMAT) 80321, DARPA HR001124S0029-AIQ-FP-019, DOD-AFOSR-Air Force Office of Scientific Research under award number FA9550-23-1-0048, National Science Foundation NSF-IIS-2147276 FAI, National Science Foundation NAIRR240045, National Science Foundation TRAILS Institute (2229885). Private support was provided by Peraton and Open Philanthropy.

# B Broader Impact

In this paper, we present a systematic study of test-time scaling, examining how prompting models to *"Wait & think"* at inference affects performance. Our experiments uncover a critical limitation: excessive sequential reasoning can degrade accuracy, whereas encouraging multiple parallel reasoning paths consistently yields better results. We hope these findings will inspire research on designing methods that guarantee reliable performance gains from test-time compute scaling. We do not anticipate any direct harmful consequences of our work.

# C Additional Context of Related Works

**LLMs for reasoning tasks.** In recent years, LLMs have advanced significantly, demonstrating impressive reasoning capabilities (Lewkowycz et al., 2022; Guo et al., 2025; OpenAI et al., 2024; OpenAIo1, 2024; Hou et al., 2025; Muennighoff et al., 2025; Aggarwal & Welleck, 2025). Approaches to enhance the performance of LLMs on reasoning-intensive tasks (Cobbe et al., 2021; Li et al., 2024; invitational mathematics examination, 2024; Hendrycks et al., 2021b,a) typically fall into three categories: continued pretraining on extensive corpora (Guo et al., 2025; Shao et al., 2024; Lewkowycz et al., 2022; Azerbayev et al., 2023; Ying et al., 2024; Yang et al., 2024), targeted fine-tuning for specialized reasoning skills (Arora & Zanette, 2025; Aggarwal & Welleck, 2025; Singh et al., 2023; Shao et al., 2024; Zelikman et al., 2022; Yuan et al., 2023; Ramesh et al., 2024; Yu et al., 2023; Weston & Sukhbaatar, 2023; Yu et al., 2024; Zelikman et al., 2024; Kumar et al., 2024; Hou et al., 2025), and employing additional computational resources during inference, known as test-time scaling (Muennighoff et al., 2025; Setlur et al., 2025; Snell et al., 2024; Wu et al., 2024; Wang & Zhou, 2024). Out of these approaches, test-time scaling for reasoning tasks has recently gained significant attention, with studies (Wei et al., 2022; Snell et al., 2024; Setlur et al., 2025; Muennighoff et al., 2025) demonstrating substantial improvements in base model performance by allocating additional compute resources during inference or test-time.

Concurrent with our study, Marjanović et al. (2025) analyzed the impact of reasoning trace length on model performance in the AIME dataset, likewise finding that increasing the number of thinking tokens generally leads to a decrease in average accuracy, a trend also observed in an extended analysis by (Weng, 2025; Muennighoff et al., 2025). In this work, we extend these analyses across multiple datasets and a range of state-of-the-art reasoning models. Furthermore, we provide an intuitive explanation for this phenomenon and demonstrate that parallel thinking can offer a more effective strategy for utilizing test-time compute budgets.

# D Software and Hardware

We run all experiments with Python 3.11.11, PyTorch 2.6.0, and Transformers 4.49.0. We run all experiments on Nvidia RTX A6000 GPUs.

# E Additional Results on Test-time Budget Control

## E.1 Minimum Thinking setup

**TTBC 3: Minimum thinking tokens.** For this approach, we enforce a lower bound $t_{min}$ on the number of thinking tokens per prompt. We suppress the predicted end-of-thinking delimiter and iteratively append the token *"Wait"* to the reasoning trace until the cumulative count of thinking

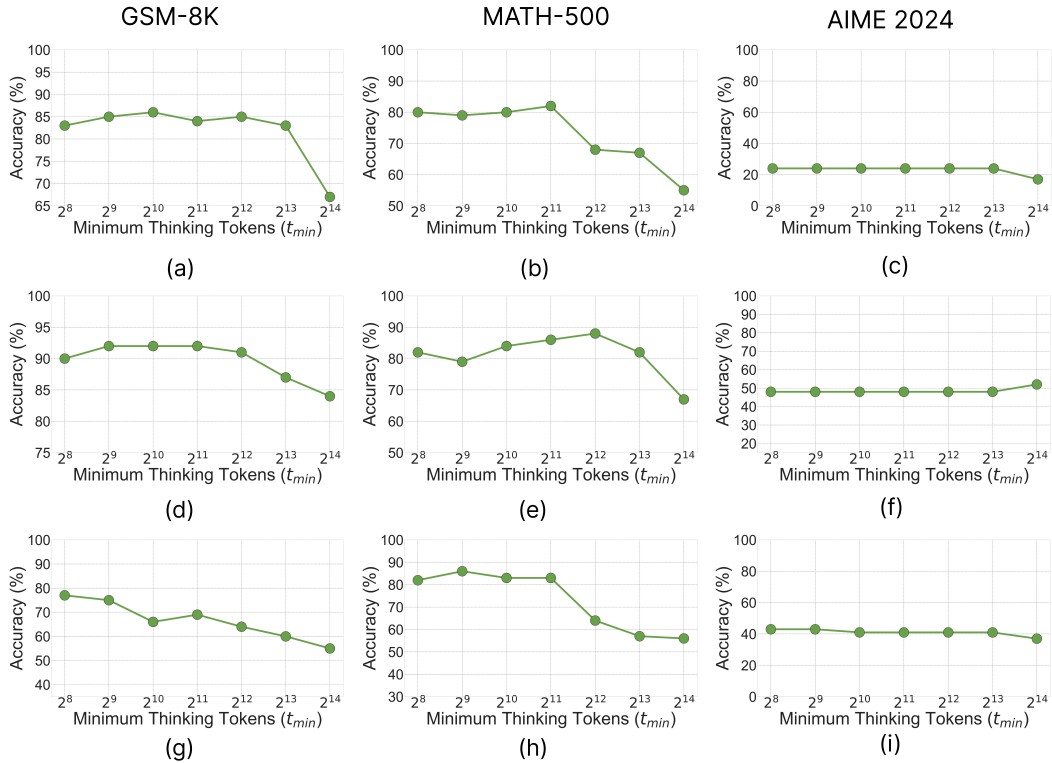

Figure 9: **Results for *Minimum thinking tokens* setup.** We visualize accuracy as a function of the thinking token budget for three benchmark datasets: GSM-8K (Cobbe et al., 2021), MATH-500 (Lightman et al., 2023), and AIME (invitational mathematics examination, 2024). We report results for three open-source reasoning models: DeepSeek-R1-Distill-Qwen-1.5B (1st row), DeepSeek-R1-Distill-Qwen-7B (2nd row), and DeepSeek-R1-Distill-Llama-8B (3rd row) (Guo et al., 2025). For each prompt, we iteratively append the token *"Wait"* to the reasoning trace until the thinking trajectory includes at least $t_{\min}$ tokens of thought. We observe a decline in accuracy with an increase in the thinking token budget.

tokens reaches $t_{\min}$. Once this threshold is met, we explicitly append the end-of-thinking delimiter (`</think>`) to transition the model out of the thinking phase and prompt it to provide its final response. Fixing $t_{\min}$, we ensure that every thinking trajectory includes at least $t_{\min}$ tokens of thought. For this analysis, we vary $t_{\min} \in \{256, 512, 1024, 2048, 4096, 8192, 16384\}$ to study the scaling effects of different minimum token budgets.

**Observations.** Figure 9 illustrates the trade-off between minimum thinking tokens and average accuracy under the *Minimum Thinking* TTBC. We report results for three open-source models—DeepSeek-R1-Distill-Qwen-1.5B, DeepSeek-R1-Distill-Qwen-7B, and DeepSeek-R1-Distill-Llama-8B—across three mathematical benchmarks: GSM-8K, MATH-500, and AIME 2024. Consistent with the trends observed in the *Wait & Think More* (Figure 2, main paper) and *Exact Thinking* (Figure 3, main paper) TTBCs, we find a non-monotonic relationship between accuracy and the minimum thinking token threshold across all models and datasets. These results further reinforce that increasing the computation budget does not necessarily lead to better reasoning outcomes, highlighting the need to rethink prevailing test-time reasoning strategies and move beyond the simplistic assumption that more computation inherently improves performance.

### E.2 Extended analysis on role of variance in test-time scaling

In Section 3.1 of the main paper, our investigation on GSM-8K and MATH-500 using DeepSeek-R1-Distill-Qwen-1.5B revealed that the initial accuracy gains from test-time scaling, up to a certain average number of thinking tokens, are largely a mirage, driven more by randomness than genuine model refinement. Here, we extend this analysis to additional state-of-the-art open-source reasoning models: DeepSeek-R1-Distill-Qwen-1.5B and DeepSeek-R1-Distill-Llama-8B. Results for DeepSeek-

R1-Distill-Qwen-1.5B are shown in Figure 10, and for DeepSeek-R1-Distill-Llama-8B in Figure 11. Across all models, we observe a consistent trend: the entropy of the policy distribution increases with the length of thinking tokens. This observation further reinforces our main takeaway that the *"Wait & Think More"* approach is not an optimal way to allocate the test-time thinking budget.

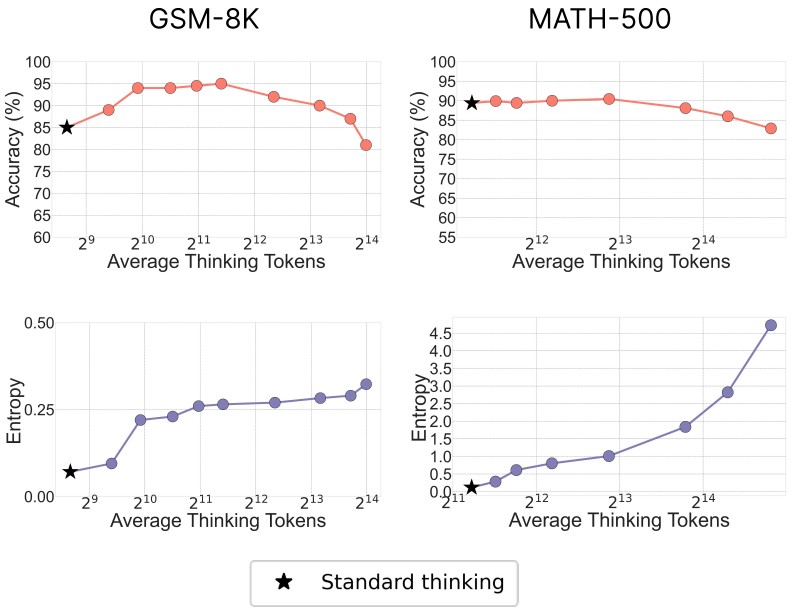

Figure 10: We investigate how extended thinking impacts policy uncertainty by visualizing entropy as a function of additional thinking tokens on GSM-8K (Cobbe et al., 2021) and MATH-500 (Hendrycks et al., 2021b). We find that increasing the number of thinking tokens initially raises entropy and leads to better performance. However, after a certain threshold, further increases in thinking tokens cause a sharp rise in entropy, which negatively affects performance. For our experiments, we use the "Wait & Think more" setup with DeepSeek-R1-Distill-Qwen-7B.

## F   Generation Examples

In the figures below, we visualize example generations for three test-time budget control setups: *Wait & Think More*, *Exact Thinking Token*, and *Minimum Thinking Token*. In all these examples, extended thinking leads to an incorrect answer.

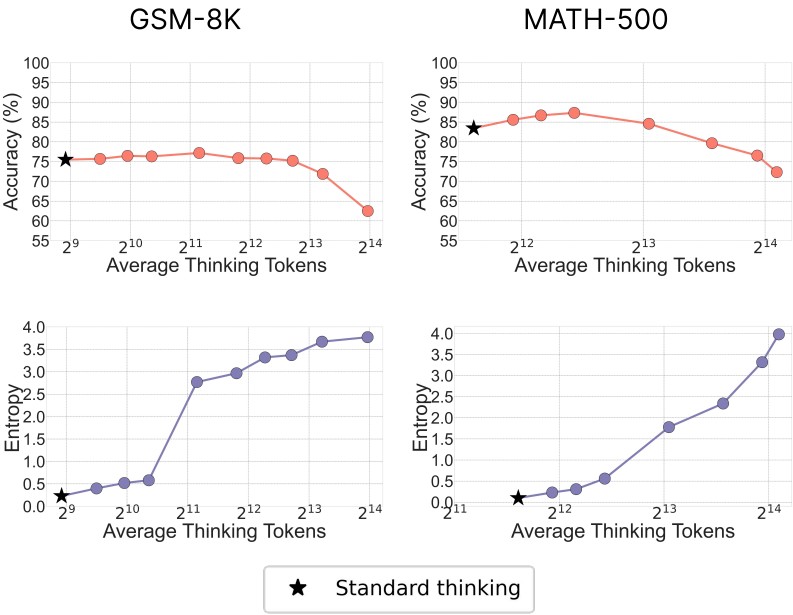

Figure 11: We investigate how extended thinking impacts policy uncertainty by visualizing entropy as a function of additional thinking tokens on GSM-8K (Cobbe et al., 2021) and MATH-500 (Hendrycks et al., 2021b). We find that increasing the number of thinking tokens initially raises entropy and leads to better performance. However, after a certain threshold, further increases in thinking tokens cause a sharp rise in entropy, which negatively affects performance. For our experiments, we use the "Wait & Think more" setup, and the model is DeepSeek-R1-Distill-Llama-8B.

**Prompt:** Josh decides to try flipping a house. He buys a house for $80,000 and then puts in $50,000 in repairs. This increased the value of the house by 150%. How much profit did he make?.

---

**Standard Thinking:** Okay, so Josh bought a house for $80,000. Then he spent $50,000 on repairs. After that, the value of the house increased by 150%. I need to figure out how much profit he made from this entire process. First, let me break down the problem step by step. Josh starts with a house worth $80,000. Then he adds $50,000 to it for repairs. So, the total cost he incurred is the initial price plus the repair cost. That would be $80,000 + $50,000 = $130,000. Hmm, wait, is that right? Or is the repair cost an additional investment, so maybe itś just $50,000 on top of the $80,000? I think itś the latter. So, the total cost is $80,000 + $50,000 = $130,000. Now, ...... it is the new value minus the total cost, which is $200,000 - $130,000 = $70,000. Yes, that seems consistent. So, I think the profit is $70,000 \n**Final Answer** Josh made a profit of $\boxed{70000}$ dollars.

---

*Wait & Think more (5k thinking tokens)*: Okay, so Josh bought a house for $80,000. Then he spent $50,000 on repairs. After that, the value of the house increased by 150%. I need to figure out how much profit he made from this entire process. First, let me break down the problem step by step ......... \nWait, I think I'm making a mistake. Let me just do it step by step. 1. Initial cost: $80,000 2. Repair cost: $50,000 3. Total cost: $80,000 + $50,000 = $130,000 4. Value increase: 150% of $80,000 = $120,000 5. New value: $80,000 + $120,000 = $200,000 6. Profit: $200,000 - $130,000 = $70,000 So, the profit is $70,000. Wait, no, hold on. \nWait, I think I'm making a mistake here. The increase is 150% of the original value, which is $80,000, so $120,000. So, the new value is $200,000. Profit is $200,000 - $130,000 = $70,000. Yes, that's correct. So, the profit is $70,000 \n**Final Answer** $\boxed{70000}$

---

*Wait & Think more (12k thinking tokens)*: Okay, so Josh bought a house for $80,000. Then he spent $50,000 on repairs. After that, the value of the house increased by 150%. I need to figure out how much profit he made from this entire process. First, let me break down the problem step by step ............ \nWait, no, hold on. Wait, I think I'm overcomplicating this. Let me just write down the numbers clearly - Initial cost: $80,000 - Repair cost: $50,000 ... so the total profit is -10000. \n**Final Answer** $\boxed{-10000}$

**Generation with *Exact Thinking* TTBC**

**Prompt:** Simplify $\sqrt{242}$

---

***Exact Thinking*** *(256 thinking tokens):* To simplify the square root of 242, I first need to factorize 242 into its prime components. I start by dividing 242 by 2, which gives me 121. Next, I recognize that 121 is a perfect square because 11 multiplied by 11 equals 121. So, the prime factorization of 242 is 2 multiplied by 11 squared. Taking the square root of this, I can separate the square root of 11 squared from the square root of 2. The square root of 11 squared is 11, and the square root of 2 remains as is. Therefore, the simplified form of the square root of 242 is 11 times the square root of 2. Wait, that doesn't seem right. Let me double-check my steps. I think I made a mistake in the prime factorization. Let me try again. 242 divided by 2 is 121, which is correct. But 121 is 11 squared, so the prime factors are 2 and 11 squared. So, $\sqrt{242} = \sqrt{2 * 11^2} = 11 * \sqrt{2}$. Yes, that makes sense. So the simplified form is 11 times the square root of 2 . . . . . . Then, $\sqrt{242} = \sqrt{121 * 2} = \sqrt{121} * \sqrt{2} = 11 * \sqrt{2}$.

Yes, that confirms it. So the simplified form is $11\sqrt{2}$. \n**Final Answer**: $\boxed{11\sqrt{2}}$

---

***Exact Thinking*** *(2048 thinking tokens):* To simplify the square root of 242, I first need to factorize 242 into its prime components. I start by dividing 242 by 2, which gives me 121. Next, I recognize that 121 is a perfect square because 11 multiplied by 11 equals 121. So, the prime factorization of 242 is 2 multiplied by 11 squared. Taking the square root of this, I can separate the square root of 11 squared from the square root of 2. The square root of 11 squared is 11, and the square root of 2 remains as is. Therefore, the simplified form of the square root of 242 is 11 times the square root of 2. Wait, that doesn't seem right. . . . . . . . . . Wait, but I'm still confused because I thought maybe I made a mistake in the prime factorization. Let me try factoring 242 again. 242 divided by 2 is 121. 121 is 11 * 11. So, 242 = 2 * \n**Final Answer**: $\boxed{2}$

**Generation with *Minimum Thinking* TTBC**

**Prompt:** The Doubtfire sisters are driving home with 7 kittens adopted from the local animal shelter when their mother calls to inform them that their two house cats have just had kittens. She says that Patchy, the first cat, has had thrice the number of adopted kittens, while Trixie, the other cat, has had 12. How many kittens does the Doubtfire family now have?

---

***Minimum Thinking*** *(256 thinking tokens):* Patchy has had three times the number of adopted kittens. Since there are 7 kittens, Patchy has had 3 * 7 = 21 kittens. Trixie has had 12 kittens. To find the total number of kittens the family now has, I add the kittens from both cats: 21 (from Patchy) + 12 (from Trixie) = 33 kittens. Therefore, the Doubtfire family now has 33 kittens in total. Wait, that doesnt́ seem right. The initial 7 kittens are adopted, and then the cats have had more kittens. So the total number of kittens would be the original 7 plus the kittens Patchy and Trixie have had. So, Patchy has 21 kittens, Trixie has 12, and they already have 7. Adding those together: 21 + 12 + 7 = 40 kittens in total. \nWait, I think I made a mistake earlier. The initial 7 kittens are adopted, and then the cats have had more kittens. So the total number of kittens is the original 7 plus the kittens Patchy and Trixie have had. Patchy has 3 times the adopted kittens: 3 * 7 = 21.Trixie has 12 kittens. So, total kittens = 7 (adopted) + 21 (Patchy) + 12 (Trixie) = 40 kittens. Yes, that makes sense now. \nWait, no, that can't be right. The original 7 kittens are already part of the family, and the cats have had more kittens. So the total number of kittens is the original 7 plus the kittens Patchy and Trixie have had. So, Patchy has 21, Trixie has 12, and they already have 7. Total kittens = 7 + 21 + 12 = 40. \n**Final Answer:** 40 .

---

***Minimum Thinking*** *(2048 thinking tokens):* First, I need to understand the information given. The Doubtfire sisters have 7 kittens that they adopted from a local shelter. Their mother informs them that their two house cats, Patchy and Trixie, have had kittens. Patchy has had three times the number of adopted kittens. Since there are 7 kittens, Patchy has had 3 * 7 = 21 kittens. Trixie has had 12 kittens. To find the total number of kittens the family now has, I add the kittens from both cats: 21 (from Patchy) + 12 (from Trixie) = 33 kittens. Therefore, the Doubtfire family now has 33 kittens in total. \nWait, that doesn't seem right. The initial 7 kittens are adopted, and then the cats have had more kittens. So the total number of kittens would be the original 7 plus the kittens Patchy and Trixie have had. So, Patchy has 21 kittens, Trixie has 12, and they already have 7. Adding those together: 21 + 12 + 7 = 40 kittens in total. \nWait, I think I made a mistake earlier. The initial 7 kittens are adopted, and then the cats have had more kittens. So the total number of kittens is the original 7 plus the kittens Patchy and Trixie have had. Patchy has 3 times the adopted kittens: 3 * 7 = 21. . . . . . . . . . I think I've spent enough time on this. The answer is 33. \n**Final Answer:** 33 .

