# OpenReview forum: "Does Thinking More Always Help? Mirage of Test-Time Scaling in Reasoning Models"
_NeurIPS.cc/2025/Conference — NeurIPS 2025 poster_

### Official Review · Reviewer_dgbk · 2025-06-29

**Clarity:** 3
**Significance:** 2
**Originality:** 2
**Rating:** 5
**Confidence:** 3

**Summary:**

This paper investigates whether increasing the amount of reasoning tokens (i.e., length of the model's chain-of-thought generation) at test time improves performance in large language models. The authors empirically analyze the effects of extended thinking on model accuracy and output entropy across three reasoning benchmarks: GSM-8K, MATH-500, and AIME 2024.

The study finds that while moderate increases in chain length initially improve accuracy, excessively long reasoning traces can lead to higher output entropy and degraded performance. The authors attribute this to growing variance in the model’s output distribution.

To address this, the paper proposes a "parallel thinking" strategy, which samples multiple reasoning paths and aggregates answers through majority voting. This approach is shown to be more robust to overthinking and maintains or improves accuracy as the thinking budget increases.

**Key Contributions:**
- Empirical evidence that longer chain-of-thoughts can hurt model accuracy due to increased output entropy.
- Comparative analysis of sequential vs. parallel test-time reasoning strategies.
- Demonstration that majority voting over multiple reasoning paths can mitigate overthinking effects.

**Questions:**

- How many parallel chains of thought are generated? Is it a fixed or flexible number (based on the total budget)?
- If it is flexible, how do the authors decide each CoT's token length? Is it the measured optimal length? If so, how to ensure the setting generalizes in a novel, even without a ground truth scenario?

**Ethical Concerns:**

["NO or VERY MINOR ethics concerns only"]

**Final Justification:**

The authors have provided two results to confirm that the main diagnosis of overthinking is not an issue of the long in-context problems, but the reasoning itself. Also, the authors added an analysis of difficulty, showing that the turning point of overthinking does vary across different tasks-easier tasks are more early to occur overthinking, while difficult tasks occur later. This result adds to the understanding of the overthinking phenomenon and its properties.

Given above, I am convinced this paper is a stronger empirical one and would love to increase my score to 5 to have clear acceptance.

**Limitations:**

Yes.

**Paper Formatting Concerns:**

No formatting issues are found.

**Quality:**

2

**Strengths And Weaknesses:**

**Strengths:**

- **Clear empirical findings.** The paper demonstrates the effects of overthinking with well-designed experiments and clear manipulations of reasoning length.
- **Intermediate explanation.** The authors use a probabilistic framework and probe the entropy of the final answer to explain why longer chains of thought may degrade performance—namely, due to increased entropy.
- **Practical solution.** The authors propose a simple yet effective method to mitigate the issue: splitting a long reasoning trace into multiple, independently sampled chains of thought and using majority voting to aggregate the results. This preserves accuracy while keeping each reasoning path within a manageable length.

**Weaknesses:**

- **Ambiguity in the notion of “thinking more.”** The title *“Does thinking more always help?”* is potentially misleading. In practice, thinking more could mean reasoning with greater diversity, exploring deeper cognitive paths, or acquiring more relevant information—not just generating longer token sequences. The paper focuses solely on the “resource” aspect (token length) of reasoning, which narrows the scope significantly and may overclaim its implications.
- **Superficial explanation of degradation.** While the authors provide an entropy-based explanation for why longer reasoning hurts performance, the analysis may not go deep enough. The phenomenon of performance drop with longer input could reflect a broader issue in LLMs—such as difficulty managing long contexts—rather than being unique to reasoning. Prior work shows that LLMs tend to become more uncertain as context length increases. It would be more convincing to disentangle whether the degradation is due to reasoning difficulty or general limitations in long-context processing. A potential experiment could be to append repeated short chains to reach a longer sequence length. If entropy still rises and performance drops despite no new information, it would suggest that the issue lies in context accumulation, not reasoning per se.
- **Lack of difficulty analysis.** The paper does not investigate whether the optimal reasoning length varies across different problem difficulties. It would strengthen the findings to show, for instance, that more complex problems require longer reasoning traces, while simple problems suffer from overthinking. This would make the core effect more interpretable.
- **Limited originality in proposed solution.** The parallel sampling and majority voting method is not novel. Similar ideas have been proposed in earlier work, most notably in Wang et al. (2022), who introduced the *self-consistency* method. Their approach samples multiple reasoning paths and aggregates answers via majority voting. The current paper does not cite or acknowledge this, which may undermine the originality of its solution.

**References:**

- Liu, N. F., Lin, K., Hewitt, J., Paranjape, A., Bevilacqua, M., Petroni, F., & Liang, P. (2023). *Lost in the middle: How language models use long contexts.* arXiv preprint arXiv:2307.03172.
- Braverman, M., Chen, X., Kakade, S., Narasimhan, K., Zhang, C., & Zhang, Y. (2020). *Calibration, entropy rates, and memory in language models.* In *International Conference on Machine Learning* (pp. 1089–1099). PMLR.
- Wang, X., Wei, J., Schuurmans, D., Le, Q., Chi, E., Narang, S., ... & Zhou, D. (2022). *Self-consistency improves chain of thought reasoning in language models.* arXiv preprint arXiv:2203.11171.

---

> ### Author Rebuttal · Authors · 2025-07-31
>
> We thank the reviewer for their thoughtful feedback and for highlighting the strengths of our work. We are encouraged that you found our paper to have **clear empirical findings**, an **intermediate explanation** for performance degradation based on entropy, and a **practical solution** to the problem. We address the specific points you raised below.
>
> > **Weakness 1:** Ambiguity in the notion of “thinking more.” The title “Does thinking more always help?” is potentially misleading. In practice, thinking more could mean reasoning with greater diversity, exploring deeper cognitive paths, or acquiring more relevant information, not just generating longer token sequences. The paper focuses solely on the “resource” aspect (token length) of reasoning, which narrows the scope significantly and may overclaim its implications.
>
>
> **Response to Weakness 1** Thank you for your points. Here are the detailed pointwise responses.
>
> - **Clarifying what “thinking more” means.** In our work, “thinking more” refers to longer chain-of‑thought reasoning at test time, the variant explored in recent literature. Our experiments reveal a non‑monotonic “overthinking” curve; accuracy improves with the first few extensions, but then degrades as the chain grows. This degradation stems from increased variance in the model’s output when reasoning is excessively prolonged. This finding is novel and highlights that, under a fixed budget, sequentially generating longer traces is unreliable.
>
> - **Addressing diversity in reasoning paths.**  We agree that “thinking more” can also mean sampling more diverse solutions or retrieving additional information. In that regard, ***our proposed parallel thinking implicitly encourages diversity***: instead of spending the entire budget on one chain, it divides the budget across multiple shorter chains and aggregates their answers by majority vote. This strategy distributes compute across diverse reasoning paths and yields higher accuracy and stability. We will revise the manuscript to clarify these different notions of “more thinking” and to emphasise that parallel thinking addresses the diversity dimension as well as the length dimension.
>
> > **Weakness 2:** Superficial explanation of degradation. While the authors provide an entropy-based explanation for why longer reasoning hurts performance, the analysis may not go deep enough. The phenomenon of performance drop with longer input could reflect a broader issue in LLMs—such as difficulty managing long contexts—rather than being unique to reasoning.
>
> **Response to Weakness 2.** Thank you for this insightful question. As suggested by the reviewer, **we performed a new ablation** (with results in Table 7 below) where we increased the length of the reasoning trace by simply repeating the same reasoning token $z \sim \pi(\cdot|x)$ multiple times and then observed the entropy of the resulting response distribution.
>
> **Table 7: Support of our explanations in the paper.** We observed that entropy does not increase just by increasing the length of the input with repeated reasoning, supporting our claim that it is **not just the context length but the reasoning** that affects the performance.
>
>
> | Average Thinking Tokens | Entropy (across response y) | Accuracy (in %) |
> |-------------------------|---------|-----------------|
> | 384.8                   | 0.245   | 82.3            |
> | 768.9                   | 0.243   | 82.3            |
> | 1153.2                  | 0.248   | 82.3            |
> | 2308.8                  | 0.237   | 82.3            |
> | 3848.1                  | 0.238   | 82.3            |
> | 4617.6                  | 0.246   | 82.3            |
> | 5771.9                  | 0.246   | 82.3            |
> | 9235.2                  | 0.241   | 82.3            |
>
>
> The model exhibits high diversity across reasoning traces, but the corresponding response is nearly deterministic for any fixed trace. Therefore, merely extending a single reasoning path does not increase entropy, as shown in the above table. In contrast, sequential thinking introduces diverse reasoning traces, which in turn increase the variance of the response distribution, hence increasing entropy (Figure 5c).
>
> > **Weakness 3:** Lack of difficulty analysis. The paper does not investigate whether the optimal reasoning length varies across different problem difficulties. It would strengthen the findings to show, for instance, that more complex problems require longer reasoning traces, while simple problems suffer from overthinking. This would make the core effect more interpretable.
>
> **Response to Weakness 3**. Excellent suggestion! As suggested by the reviewer, we conducted an analysis on how the hardness of the problem affects the overthinking issue. For this experiment, we utilized the MATH-500 benchmark, which includes problems across five levels of difficulty. We report the average accuracy for problems of varying difficulty in the table below:
>
>
> **Table 8:** Accuracy across problems of varying difficulty in MATH-500 with Wait & Think More setup.
>
> | Number of times "Wait" is appended | Level-1 | Level-2 | Level-3 | Level-4 | Level-5 |
> |------------------------------------|---------|---------|---------|---------|---------|
> | 0                                  | 100     | 100     | 95.2    | 92.2    | 82.1    |
> | 1                                  | 100     | 100     | 97.1    | 92.9    | 83.1    |
> | 2                                  | 97.7    | 100     | 97.1    | 92.9    | 83.1    |
> | 3                                  | 88.4    | 93.3    | 94.3    | 92.9    | 84.3    |
> | 4                                  | 81.4    | 90.0    | 90.5    | 85.9    | 82.1    |
> | 5                                  | 74.4    | 85.6    | 88.6    | 84.3    | 80.6    |
>
>
> **Takeaway:** We observe that there is a correlation between the decline in accuracy with the difficulty of the problem. Specifically, performance on relatively easier problems (Levels 1 and 2) drops earlier as overthinking increases, compared to harder problems (Levels 4 and 5).
>
> Thanks for the suggestion. We will include this ablation in the revised version to make our analysis concrete.
>
>
> > **Weakness 4:** Limited originality in proposed solution. The parallel sampling and majority voting method is not novel. Similar ideas have been proposed in earlier work, most notably in Wang et al. (2022), who introduced the self-consistency method. Their approach samples multiple reasoning paths and aggregates answers via majority voting. The current paper does not cite or acknowledge this, which may undermine the originality of its solution.
>
> **Response to Weakness 4**:  We note that proposing parallel sampling with majority voting is not the key novelty of our work. We take this opportunity to clarify the two **distinct contributions** beyond classic self-consistency (as also acknowledged by Reviewer f39d) :
>
> 1. **Over-thinking diagnosis:** We show that, for a fixed token budget, extending a single chain eventually harms accuracy (a non-monotonic trend), an effect not shown in prior works. To the best of our knowledge, our work is the first to highlight the issue of "overthinking" in test-time scaling of reasoning models with a detailed analysis.
> 2. **Superior budget allocation strategy:** Showing that, under the same token budget $B$, distributing the budget across multiple chains yields consistently higher accuracy and stability. Concretely, for a total budget $B$ thinking tokens, sequential scaling utilizes them on a single long chain (L=B), whereas parallel thinking uses K shorter chains of length L=B/K and aggregates predictions.
>
> **Note**: We apologize for the oversight, and we will cite and discuss Self‑consistency (Wang et al., 2022) in detail in our revised version.
>
>
>
> > **Question 1:** How many parallel chains of thought are generated? Is it a fixed or flexible number (based on the total budget)?
>
> > **Question 2:** If it is flexible, how do the authors decide each CoT's token length? Is it the measured optimal length? If so, how to ensure the setting generalizes in a novel, even without a ground truth scenario?
>
> **Response to Questions 1 and 2:** The number of parallel chains of thought generated is flexible and depends on the total token budget (let’s say B tokens). Given a total budget B, for a test set, we determine the average response length over a set of prompts (let's say K tokens). Then, the number of CoTs that can be generated is approximately $\frac{B}{K}$.

---

> > ### Comment · Reviewer_dgbk · 2025-08-01
> >
> > I appreciate the authors' rebuttal and am impressed by the new results. The results provided by the authors largely enhanced the credibility of their main diagnosis as well as solutions. I would like to clearly recommend that this paper be accepted by the conference with those revisions included. I will improve my score to 5. Again, thanks to the author for the effortful rebuttal.

---

> > > ### Author Response · Authors · 2025-08-02
> > > **Thank you Reviewer dgbk for rebuttal acknowledgement and recommending acceptance**
> > >
> > > Dear Reviewer dgbk,
> > >
> > > Thank you so much for your thoughtful review and for taking the time to consider our rebuttal. Your feedback has been invaluable in helping us improve the quality of our work. We will ensure all the revisions and new results are thoroughly integrated into the final version.
> > >
> > > Thank you once again for your effort and constructive engagement.
> > >
> > > Sincerely,
> > > Authors

---

### Official Review · Reviewer_f39d · 2025-06-30

**Clarity:** 3
**Significance:** 2
**Originality:** 2
**Rating:** 5
**Confidence:** 4

**Summary:**

This submission analyzes the efficacy of test-time scaling for reasoning in large language models (LLMs) by extending their "thinking" duration. The authors conduct a systematic empirical study and find that performance does not improve monotonically. Initially, it ascends with an increase in cognitive tokens but subsequently declines, a phenomenon known as "overthinking." The submission utilizes a probabilistic model indicating that prolonged contemplation amplifies the variance of the model's output distribution, which may be advantageous at first but ultimately diminishes accuracy.  Informed by this insight, the authors advocate for "parallel thinking": the creation of multiple independent reasoning pathways and the application of a majority vote.

**Questions:**

1. The proposed "parallel thinking" coupled with majority voting is fundamentally synonymous with self-consistency.  Could you clarify the distinction?  Does the primary contribution lie in the direct, cost-effective comparison with sequential reasoning, thereby establishing it as a superior budget allocation strategy? A more explicit discussion in the paper could strengthen the positioning of this contribution.
2. Your study provides convincing evidence on models up to 8B parameters. Do you have a hypothesis on how the "overthinking" phenomenon might manifest in much larger models (e.g., QwQ-32B or DeepSeek R1)? It would be helpful to hear about any early experiments or ideas you have on this.
3. The study uses cues like "Wait" to make people think longer.  Have you thought about how the type of this prompt might affect the result?  For example, would a more structured prompt, like "Let's go over the steps again," have a different effect on the output variance and the overthinking curve than a general one?
4. The example probabilistic model assumes that the reward function is smooth and has only one peak, but the actual reward in these reasoning tasks is a sparse, binary signal.  Could you talk about how this simplification might make it harder to apply the model's intuition to a real-world situation?  Is the idea behind this that the model's preference landscape (the chance it gives to answers) is smoother than the external reward function?

**Ethical Concerns:**

["NO or VERY MINOR ethics concerns only"]

**Final Justification:**

This submission provides detailed discussion and experimental results to address the real-world problem. In addition, the proposed method is simple and effective. In summary, this submission is worth being presented at the conference. Thus, I recommend accepting it.

**Limitations:**

yes

**Quality:**

3

**Strengths And Weaknesses:**

**Strengths:**

1. The paper talks about a real-world problem with using reasoning models.  Many tests on three different datasets and three different open-source models with different parameter scales support the main idea of "overthinking."  Figs. 2 and 3 clearly and consistently show the non-monotonic performance trend.
2. The reason for overthinking based on variance makes sense and works.   The "parallel thinking" method that is suggested is a simple but helpful way for practitioners to get clear advice on how to best use their inference budget.
3. The paper's writing is mostly clear and easy to follow.   The authors make it clear what their hypothesis, methods, and results are.   The numbers, in particular, do a good job of showing the main results.

**Weaknesses:**

1. The main weakness is that the proposed "parallel thinking" method is functionally similar to well-established techniques like self-consistency sampling. The submission frames it as an effective way to allocate a budget compared to sequential thinking, but it could be more clearly placed in the context of previous work to better show what it adds.
2. The tests are done on small models with up to 8 billion parameters. While the results are consistent within this scope, it remains an open question whether the "overthinking" phenomenon and the relative benefits of parallel thinking generalize to much larger state-of-the-art models (e.g., QwQ-32B, DeepSeekR1, or OpenAI o3). The authors acknowledge this limitation in Section 6.
3. The probabilistic model used to explain the phenomenon is a helpful illustration, but does not constitute a formal theoretical proof. It simplifies both the model's output and the reward structure. This is noted as a limitation by the authors.

---

> ### Author Rebuttal · Authors · 2025-07-31
>
> We thank you for your detailed and insightful review. We are pleased that you recognized our work addresses a **real-world problem**, is supported by **strong empirical evidence**, and that our proposed "parallel thinking" method offers **simple but helpful** advice for practitioners. We also appreciate your positive comments on the **clarity of our writing and figures**. We address the questions and weaknesses below.
>
> > **Weakness 1:** The main weakness is that the proposed "parallel thinking" method is functionally similar to well-established techniques like self-consistency sampling. The submission frames it as an effective way to allocate a budget compared to sequential thinking, but it could be more clearly placed in the context of previous work to better show what it adds.
>
> > **Question 1:** The proposed "parallel thinking" coupled with majority voting is fundamentally synonymous with self-consistency. Could you clarify the distinction? Does the primary contribution lie in the direct, cost-effective comparison with sequential reasoning, thereby establishing it as a superior budget allocation strategy? A more explicit discussion in the paper could strengthen the positioning of this contribution.
>
> **Response to Weakness 1 / Question 1:** Thank you for your insightful questions. Your impression is correct that we are establishing a superior budget allocation strategy with the help of "parallel thinking", but there is another key novelty of our work. We take this opportunity to clarify the two **distinct contributions** beyond classic self-consistency.
>
> 1. **Over-thinking diagnosis:** We show that, for a fixed token budget, extending a single chain eventually harms accuracy (a non-monotonic trend), an effect not shown in prior works. To the best of our knowledge, our work is the first to highlight the issue of "overthinking" in test-time scaling of reasoning models with a detailed analysis.
> 2. **Superior budget allocation strategy:** Showing that, under the same token budget $B$, distributing the budget across multiple chains yields consistently higher accuracy and stability. Concretely, for a total budget $B$ thinking tokens, sequential scaling utilizes them on a single long chain (L=B), whereas parallel thinking uses K shorter chains of length L=B/K and aggregates predictions.
>
> > **Weakness 2/Question 2:** Your study provides convincing evidence on models up to 8B parameters. Do you have a hypothesis on how the "overthinking" phenomenon might manifest in much larger models (e.g., QwQ-32B or DeepSeek R1)? It would be helpful to hear about any early experiments or ideas you have on this.
>
> **Response to Weakness 2/Question 2:** Thank you for this important point. We have conducted additional experiments with larger models such as Qwen-3-32B on both GSM8K (Tables 1,2) and MATH-500 (Tables 3,4). Our findings reveal a similar accuracy curve: performance initially increases with extended thinking but subsequently declines, which confirms the presence of overthinking at larger scales as well. We report the results below:
>
> **Results on GSM-8K (Table 1 and Table 2):**
>
>
> **Table 1:** Wait & Think More Setup.
>
> | Average Thinking Tokens | Wait & Think Accuracy (%) |
> |------------------------|--------------------------|
> | 1182.4  | 93.25 (Standard Thinking) |
> | 1771.5  | 96.06 (↑) |
> | 2077.1  | 96.36 (↑) |
> | 2748.6  | 94.01 (↓) |
> | 4338.9  | 91.96 (↓) |
> | 5125.3  | 85.44 (↓) |
> | 7377.7  | 83.03 (↓) |
> | 9379.7  | 82.04 (↓) |
> | 11079.1 | 79.01 (↓) |
>
>
>
> **Table 2:** Parallel Thinking Setup.
>
> | Average Thinking Tokens | Parallel Thinking Accuracy (%) |
> |------------------------|--------------------------|
> | 1182.4   | 93.25 (Standard Thinking) |
> | 2064.3   | 93.71 (↑) |
> | 3506.9   | 94.69 (↑) |
> | 5561.4   | 95.45 (↑) |
> | 7908.6   | 96.54 (↑) |
> | 11388.5  | 97.42 (↑) |
> | 16593.1  | 97.42 (↔) |
>
> **Results on MATH-500 (Table 3 and Table 4):**
>
>
> **Table 3:** Wait & Think More Setup.
>
> | Average Thinking Tokens | Wait & Think Accuracy (%) |
> |------------------------|--------------------------|
> | 2897.2   | 92.2 (Standard Thinking) |
> | 4364.3   | 93.2 (↑) |
> | 6218.5   | 93.2 (↔) |
> | 7784.3   | 90.6 (↓) |
> | 9912.6   | 89.2 (↓) |
> | 12673.3  | 86.2 (↓) |
> | 14592.8  | 83.6 (↓) |
>
>
>
> **Table 4:** Parallel Thinking Setup.
>
> | Average Thinking Tokens | Parallel Thinking Accuracy (%) |
> |------------------------|--------------------------|
> | 2897.2   | 92.2 (Standard Thinking) |
> | 5441.9   | 93.2 (↑) |
> | 8230.1   | 93.6 (↑) |
> | 11348.8  | 93.6 (↔) |
> | 14085.6  | 94.2 (↑) |
>
> ***Takeaway:*** ***Our results persist at scale.*** Accuracy initially increases with extended thinking but subsequently declines. However, as evident from the results, with parallel thinking, we see a monotonic increase in accuracy.
>
> > **Weakness 3/Question 4:** The example probabilistic model assumes that the reward function is smooth and has only one peak, but the actual reward in these reasoning tasks is a sparse, binary signal. Could you talk about how this simplification might make it harder to apply the model's intuition to a real-world situation? Is the idea behind this that the model's preference landscape (the chance it gives to answers) is smoother than the external reward function?
>
> **Response to Weakness 3/Question 4:** Thank you for this question. We agree that, for illustration, we used a simplified toy model as a first step to analyze the behavior of sequential thinking, since this is one of the first works to formally study this issue.
> **Extending Intuition of our toy example to real-world situation of sparse reward:** In sparse reward settings such as math problems where only exact answers receive high reward, the reward function is sharply peaked and zero elsewhere, which can be modeled as a Dirac delta function $r(x,y) = \delta(y -\mu_r)$ where the reward is nonzero only at a specific correct answer $\mu_r$. In this case, the expected reward becomes:
> $\mathbb{E}\_{y \sim \pi(\cdot|x)} [r(x, y)] = \int_{y} \delta(y - \mu_r) \pi(y|x) dy
> = \frac{1}{\sqrt{2\pi\sigma_\pi^2}} \exp\left(-\frac{(\mu_r - \mu_\pi)^2}{2\sigma_\pi^2}\right).$
>
> In this form, the exponential term becomes less sensitive as $\sigma^2_{\pi}$ increases while the prefactor $\frac{1}{\sqrt{2\pi\sigma_\pi^2}}$ starts to dominate and decays rapidly. Because the reward peak is so narrow, the policy has to increase its variance substantially before achieving meaningful overlap, so the expected reward increases slowly at first. But once overlap is sufficient, any further increase in $\sigma_\pi^2$ causes the prefactor to shrink faster than gains from the exponential term, resulting in a sharp drop in expected reward, which makes the overthinking curve particularly steep.
>
> > **Question 3:** The study uses cues like "Wait" to make people think longer. Have you thought about how the type of this prompt might affect the result? For example, would a more structured prompt, like "Let's go over the steps again," have a different effect on the output variance and the overthinking curve than a general one?
>
> **Response to Question 3:** Thank you for raising this point. To determine whether the specific wording of the “wait” cue matters, we replaced “Wait” with a variety of alternative prompts, including “Think again,” “Try again,” “Improve further,” and even more structured phrases such as “Let’s rethink step by step.” Across the models and datasets we studied, the resulting accuracy curves were essentially unchanged: a small gain from the first additional thinking step was invariably followed by a decline as more thinking was forced. In other words, the non‑monotonic overthinking phenomenon we report is robust to the choice of cue.
>
> The reason, as discussed in the paper, is that any instruction to continue thinking conditions the model on its previous reasoning. This increases the variance of the model’s output distribution, which initially allows occasional corrections but ultimately spreads mass over many incorrect answers. Thus, it is the act of prolonging reasoning, not any particular keyword, that drives the observed behaviour. We will include the ablation curves for the alternate prompts in the appendix of the revised version to make this explicit.
>
>
> We include the ablations below in Tables 5 and 6 on GSM-8K using Deepseek-Distill-Qwen-1.5B, and will add them to the appendix in the updated version.
>
> **Table 5:** Results using "Think again" on GSM-8K.
>
> |Average Thinking Tokens|Wait & Think Accuracy(%)|
> |----------------------|-----------|
> |384.8|82.3 (Standard Thinking)|
> |582.3|84.5 (↑)|
> |1085.1|86.3 (↑)|
> |2766.9|86.3 (↔)|
> |3417.9|87.2 (↑)|
> |6931.5|82.9 (↓)|
> |9318.2|75.0 (↓) |
> |12980.3|71.6 (↓)|
>
>
> **Table 6:** Results using "Let’s rethink step by step" on GSM-8K.
>
>
> |Average Thinking Tokens|Wait & Think Accuracy(%)|
> |----------------------|-----------|
> |384.8|82.3 (Standard Thinking)|
> |596.1|84.4 (↑)|
> |1198.3|86.4 (↑)|
> |2512.2|86.5 (↔)|
> |3804.7|87.3 (↑)|
> |7311.6|82.7 (↓)|
> |9486.5|75.4 (↓) |
> |13120.9|74.4 (↓)|

---

> > ### Comment · Reviewer_f39d · 2025-08-02
> >
> > Thank you for providing a detailed discussion and additional experiment results. All my concerns are well-addressed. I will raise my score from 4 to 5.

---

> > > ### Author Response · Authors · 2025-08-02
> > > **Thank you Reviewer f39d for rebuttal acknowledgement and recommending acceptance**
> > >
> > > Dear Reviewer f39d,
> > >
> > > Thank you for your effort and time in reviewing our work. Your feedback has helped us improve the work. Thank you once again for recommending the acceptance of our work.
> > >
> > > Regards,
> > > Authors

---

### Official Review · Reviewer_WH7z · 2025-07-01

**Clarity:** 2
**Significance:** 2
**Originality:** 3
**Rating:** 4
**Confidence:** 5

**Summary:**

This paper questions whether the performance gains from test-time scaling continue to improve indefinitely as more “thinking” (i.e., sequential computation) is applied. To this end, the paper uses some tricks to force the model to think more (e.g., "Wait"), and observes that the model's performance degrades at some point. Thus the author propose joint usage of parallel thinking is effective.

**Questions:**

I think it is quite interesting that test-time scaling works for somewhat relatively easy datasets (e.g., GSM-8K, MATH). Do the authors have any explanation why test-time scaling does not work for hard datasets such as AIME?

**Ethical Concerns:**

["NO or VERY MINOR ethics concerns only"]

**Final Justification:**

I think the paper has the strength to be accepted. The major concerns of the paper were well-addressed through the rebuttal.

**Limitations:**

See the weakness above.

**Paper Formatting Concerns:**

No concern.

**Quality:**

3

**Strengths And Weaknesses:**

**Strength**

(1) Well-motivated and great topic that should be explored

(2) Overall analysis is well conducted

(3) The writing is clear

**Weakness**

(1) I have a concern as it is an out-of-distribution scenario. While I do agree that S1 [1] has shown "Wait & Think more" is effective for test-time scaling, we are not sure that this is an optimal way for scaling. Specifically, the R1 model family is not trained to scale based on that condition; rather, it is just an interesting finding (e.g., we don't know whether this works for OpenAI o-series model or other models as well).

(2) I think it would be great if the authors could consider more model families than R1, e.g., QwQ-32B. Since this paper is an analysis paper, it should consider a broad range of model families. I think it is worth reporting even if the trends are inconsistent from R1.

(3) I think we still need test-time-scaling that is non-parallel if the answer is hard to aggregate. For example, coding [2,3] or summarizing problems makes it hard to use parallel test-time-scaling. Therefore, I think this may limit the scope of the paper.

(4) Extending the question from (3), I think it is worth investigating the test-time-scaling trend in coding [2,3] datasets where parallel thinking is hard to apply.

(5) Considering more math benchmarks are required. This paper is mainly about analysis. However, the paper has only considered three datasets. Considering some benchmarks like AIME 2025, AMC2023, and GPQA Diamond is encouraged.

------------------

Overall, I think the paper has strengths to be accepted. If the concerns are well-addressed, I am willing to increase the score.

[1] s1: Simple test-time scaling\
[2] Program Synthesis with Large Language Models\
[3] Evaluating Large Language Models Trained on Code

---

> ### Author Rebuttal · Authors · 2025-07-31
>
> We sincerely thank you for your constructive comments and positive assessment. We are glad you found the paper **well-motivated** with a **great topic**, the analysis **well-conducted**, and the writing **clear**. We appreciate that you see the paper's **strengths to be accepted** and have addressed the concerns below to further strengthen our work.
>
> > **Weakness 1:** I have a concern as it is an out-of-distribution scenario. While I do agree that S1 [1] has shown "Wait & Think more" is effective for test-time scaling, we are not sure that this is an optimal way for scaling. Specifically, the R1 model family is not trained to scale based on that condition; rather, it is just an interesting finding (e.g., we don't know whether this works for OpenAI o-series model or other models as well).
>
> > **Weakness 2:** I think it would be great if the authors could consider more model families than R1, e.g., QwQ-32B. Since this paper is an analysis paper, it should consider a broad range of model families. I think it is worth reporting even if the trends are inconsistent from R1.
>
> **Response to Weakness 1-2.** This is an interesting point. We have extended our analysis to new model families (Qwen-3-32B, as suggested by the reviewers) and report ***similar trends for larger models as well*** on GSM-8K in Tables 1, 2, and MATH-500 in Tables 3, 4  (we are committed to include more large-scale experiments for the final version):
>
>
> **Results on GSM-8K (Table 1 and Table 2):**
>
>
> **Table 1:** Wait & Think More Setup.
>
> | Average Thinking Tokens | Wait & Think Accuracy (%) |
> |------------------------|--------------------------|
> | 1182.4  | 93.25 (Standard Thinking) |
> | 1771.5  | 96.06 (↑) |
> | 2077.1  | 96.36 (↑) |
> | 2748.6  | 94.01 (↓) |
> | 4338.9  | 91.96 (↓) |
> | 5125.3  | 85.44 (↓) |
> | 7377.7  | 83.03 (↓) |
> | 9379.7  | 82.04 (↓) |
> | 11079.1 | 79.01 (↓) |
>
>
>
> **Table 2:** Parallel Thinking Setup.
>
> | Average Thinking Tokens | Parallel Thinking Accuracy (%) |
> |------------------------|--------------------------|
> | 1182.4   | 93.25 (Standard Thinking) |
> | 2064.3   | 93.71 (↑) |
> | 3506.9   | 94.69 (↑) |
> | 5561.4   | 95.45 (↑) |
> | 7908.6   | 96.54 (↑) |
> | 11388.5  | 97.42 (↑) |
> | 16593.1  | 97.42 (↔) |
>
>
> **Results on MATH-500 (Table 3 and Table 4):**
>
>
> **Table 3:** Wait & Think More Setup.
>
> | Average Thinking Tokens | Wait & Think Accuracy (%) |
> |------------------------|--------------------------|
> | 2897.2   | 92.2 (Standard Thinking) |
> | 4364.3   | 93.2 (↑) |
> | 6218.5   | 93.2 (↔) |
> | 7784.3   | 90.6 (↓) |
> | 9912.6   | 89.2 (↓) |
> | 12673.3  | 86.2 (↓) |
> | 14592.8  | 83.6 (↓) |
>
>
>
> **Table 4:** Parallel Thinking Setup.
>
> | Average Thinking Tokens | Parallel Thinking Accuracy (%) |
> |------------------------|--------------------------|
> | 2897.2   | 92.2 (Standard Thinking) |
> | 5441.9   | 93.2 (↑) |
> | 8230.1   | 93.6 (↑) |
> | 11348.8  | 93.6 (↔) |
> | 14085.6  | 94.2 (↑) |
>
> > **Weakness 3:** I think we still need test-time-scaling that is non-parallel if the answer is hard to aggregate. For example, coding [2,3] or summarizing problems makes it hard to use parallel test-time-scaling. Therefore, I think this may limit the scope of the paper.
>
> > **Weakness 4:** Extending the question from (3), I think it is worth investigating the test-time-scaling trend in coding [2,3] datasets where parallel thinking is hard to apply.
>
> **Response to Weakness 3 & 4:** This is an excellent point, and we agree that in certain settings, such as code generation or summarization, majority voting over outputs can be difficult or even infeasible. However, **several effective alternatives exist for aggregating responses from parallel chains.** One effective alternative in such cases is to estimate the log-probability of each response under the model $\log \pi(y|x)$, and select the response that maximizes this score. We validate our hypothesis through experiments on a code generation task:
>
>
> **Results on Code Generation Task (Table 5 and Table 6):**
>
> We report results for the MBPP (Muennighoff/mbpp) Python code generation benchmark on Nvidia Llama-3.1-Nemotron-Nano-4B-v1.1 thinking model using different setups:
>
> **Table 5:** Wait & Think More Setup.
>
> | Average Thinking Tokens | Wait & Think Accuracy (%) |
> |------------------------|-------------------------------|
> | 3804.2 | 80.08 (Standard Thinking) |
> | 6435.1 | 78.02 (↓) |
> | 8889.4 | 78.02 (↔) |
> | 9484.3 | 74.94 (↓) |
> | 12306 | 70.42 (↓) |
>
>
>
> **Table 6:** Parallel Thinking Setup.
>
> | Average Thinking Tokens | Parallel Thinking Accuracy (%) |
> |------------------------|-------------------------------|
> | 3804.2 | 80.08 (Standard Thinking) |
> | 7262.4 | 80.08 (↔) |
> | 10941.2 | 80.08 (↔) |
> | 15862.8 | 80.18 (↑) |
> | 19010.3 | 80.18 (↔) |
>
>
> To further validate our observations, we also report findings on a multi-task reasoning benchmark:
>
> **Results on Multi-task Reasoning Task (Table 7 and Table 8):**
>
> We report results for the MMLU-Pro benchmark (TIGER-Lab/MMLU-Pro) on DeepSeek-R1-Qwen 1.5B using different setups:
>
> **Table 7:** Wait & Think More Setup.
>
> | Average Thinking Tokens | Wait & Think Accuracy (%) |
> |------------------------|--------------------------|
> | 637.2   | 30.00 (Standard Thinking) |
> | 980.0   | 34.28 (↑) |
> | 1071.6  | 38.57 (↑) |
> | 1702.9  | 35.71 (↑) |
> | 2005.7  | 35.71 (↔) |
> | 2144.5  | 44.28 (↑) |
> | 2629.2  | 41.42 (↓) |
> | 2944.9  | 40.00 (↓) |
> | 4874.4  | 34.28 (↓) |
> | 5120.5  | 30.00 (↓) |
>
>
> **Table 8:** Parallel Thinking Setup.
>
> | Average Thinking Tokens | Parallel Thinking Accuracy (%) |
> |------------------------|-------------------------------|
> | 637.2 | 30.00 (Standard Thinking) |
> | 1153.8 | 34.28 (↑) |
> | 2590.3 | 36.48 (↑) |
> | 3980.1 | 36.72 (↑) |
> | 5296.1 | 42.21 (↔) |
> | 9775.5 | 48.24 (↑) |
>
> **Takeaway:** We observe a similar trend: accuracy initially increases with extended thinking but subsequently declines. However, with parallel thinking, we see a monotonic increase in accuracy.
>
> > **Weakness 5:** Considering more math benchmarks are required. This paper is mainly about analysis. However, the paper has only considered three datasets. Considering some benchmarks like AIME 2025, AMC2023, and GPQA Diamond is encouraged.
>
> **Response to Weakness 5:** Thank you for this suggestion. We agree that broader coverage of math benchmarks strengthens the analysis. In the original submission, we had included results on AIME 2024, which aligns with our findings. Based on the reviewer's feedback, we have ***now extended our experiments to two additional high‑difficulty datasets: GPQA‑Diamond and AMC 2023, using the DeepSeek‑Qwen‑1.5B model.*** The new results, presented below, show consistent trends across all evaluated benchmarks, further supporting our hypothesis. We appreciate the opportunity to demonstrate the generality of our approach and believe these additions make the paper more comprehensive.
>
> **Results on GPQA-Diamond (Table 9 and Table 10):**
>
> **Table 9:** Wait & Think More Setup.
>
> | Average Thinking Tokens | Wait & Think Accuracy (%) |
> |------------------------|-------------|
> | 3503.6   | 33.33 (Standard Thinking) |
> | 5673.5   | 35.85 (↑) |
> | 6300.4   | 36.86 (↔) |
> | 6733.9   | 36.86 (↔) |
> | 7506.4   | 32.82 (↓) |
> | 8993.2   | 31.81 (↓) |
> | 9976.1   | 30.30 (↓) |
> | 11904.5  | 30.30 (↔) |
> | 13900.4  | 28.78 (↓) |
>
> **Table 10:** Parallel Thinking Setup.
>
> | Average Thinking Tokens | Parallel Thinking Accuracy (%) |
> |------------------------|-------------|
> | 3503.6   | 33.33 (Standard Thinking) |
> | 6988.7   | 33.83 (↑)|
> | 10081.1  |  35.85 (↑)|
> | 11665.9  |  37.87 (↑) |
> | 13787.9  | 37.87 (↔) |
>
>
> **Results on AMC 2023 (Table 11 and Table 12):**
>
> **Table 11:** Wait & Think More Setup.
>
> | Average Thinking Tokens | Wait & Think Accuracy (%) |
> |------------------------|--------------------------|
> | 2827.9   | 77.5 (↑) |
> | 3513.1   | 80.0 (↑) |
> | 4902.7   | 85.0 (↑) |
> | 5471.6   | 82.5 (↓) |
> | 6257.7   | 82.5 (↔) |
> | 6600.1   | 82.5 (↔) |
> | 8218.1   | 75.0 (↓) |
> | 9657.8   | 67.5 (↓) |
> | 10098.4  | 67.5 (↔) |
> | 14587.4  | 60.0 (↓) |
>
>
> **Table 12:** Parallel Thinking Setup.
>
> | Average Thinking Tokens | Parallel Thinking Accuracy (%) |
> |------------------------|--------------------------|
> | 2827.9   | 77.5 (↑) |
> | 5574.8   | 80.0 (↑) |
> | 7371.4   | 80.0 (↔) |
> | 9832.5   | 85.0 (↑) |
> | 12290.1   | 87.5 (↑) |
> | 14730.8  | 87.5 (↔) |
>
> > **Question:** I think it is quite interesting that test-time scaling works for somewhat relatively easy datasets (e.g., GSM-8K, MATH). Do the authors have any explanation why test-time scaling does not work for hard datasets such as AIME?
>
> **Response to Question**:  In our experiments, we have observed that test‑time scaling does provide noticeable gains on difficult AIME benchmarks, particularly for larger models. As illustrated in Fig. 6 (f, i), accuracy for DeepSeek‑R1‑Distill‑Qwen‑7B and DeepSeek‑R1‑Distill‑Llama‑8B increases from roughly 20% to around 60% when we apply test‑time scaling, and climbs even higher with parallel thinking. This demonstrates that the technique does help on AIME.
>
> That said, we agree that the relative improvement from test-time scaling on harder tasks may be less pronounced for smaller models (1.5B), possibly due to inherent capability bottlenecks that limit their performance on highly complex problems like AIME.

---

> > ### Author Response · Authors · 2025-08-04
> > **Request to Reviewer WH7z**
> >
> > Dear Reviewer WH7z,
> >
> > We sincerely thank you again for your valuable feedback, which has greatly helped us improve the quality of our work. As the discussion deadline is approaching, we kindly request you to review our rebuttal.
> >
> > We are happy to address any remaining concerns. If our responses adequately address your concerns, we kindly ask you to consider adjusting the corresponding scores. Thank you again for your time and effort!
> >
> > Regards,
> >
> > Authors

---

> > > ### Comment · Reviewer_WH7z · 2025-08-05
> > > **Thank you for the rebuttal**
> > >
> > > I thank the author for the rebuttal and for addressing my concerns with additional experiments. I believe the rebuttal has strengthened the overall claim of the paper and will increase my score accordingly.

---

### Official Review · Reviewer_3KEc · 2025-07-06

**Clarity:** 3
**Significance:** 2
**Originality:** 3
**Rating:** 4
**Confidence:** 3

**Summary:**

The authors revisit test-time “think more” prompting and discover a non-monotonic accuracy curve: after a few added “Wait/Think more” tokens, accuracy peaks and then falls—an overthinking regime that shows up on GSM-8K, MATH-500 and AIME 2024 for three open-source models (Fig. 2): DeepSeek-R1-Distill-Qwen-1.5B, DeepSeek-R1-Distill-Qwen-7B, and DeepSeek-R1-Distill-Llama-8B model.

A simple 1-D Gaussian argument plus measured entropy curves explain this mirage effect  in test-time scaling.

Motivated by this, the paper proposes parallel thinking: split the same token budget into several short, independent reasoning paths and majority-vote. Under a fixed 16 K-token budget this outperforms sequential “Wait & Think more” by ~22 pp and never drops below the peak.

**Questions:**

Please see strengths and weaknesses section.

**Ethical Concerns:**

["NO or VERY MINOR ethics concerns only"]

**Final Justification:**

Thank you authors for the rebuttal. Authors have addressed my concerns. Therefore, I increase my score to 4.

**Limitations:**

Yes, addressed in Sec. 6.

**Quality:**

2

**Strengths And Weaknesses:**

**Strengths:**

1) **Clear empirical observation:** Figure 2 consistently shows the overthinking dip across three math benchmarks and three model sizes, correcting prior “more is always better” claims.

2) **Insightful explanation**: The Gaussian overlap toy model and entropy analysis tie the phenomenon to variance growth.

3) **Simple, actionable fix**: Parallel thinking is simple yet yields large gains and avoids tuning a stopping point.


**Weaknesses:**

1) **Model scale:** Results stop at 8 B params; a single run on a 30-ish B model (e.g., Qwen-30B) would help confirm that overthinking—and the parallel fix—persist at scale.

2) **Predictive power missing:** The variance story/ schematic discussed in Sec. 3 and Fig.4 is qualitative ; it doesn’t give a formula to anticipate the overthinking peak across tasks or model scales. Without a way to estimate the tipping point in advance, practitioners still have to run experiments to find it.

3) **Decoding details:** Entropy trends depend on sampling temperature/top-p; explicitly reporting those settings, or adding a robustness table, would strengthen Section 2.

4) **Latency & throughput:** Because parallel branches can be batched, it is likely faster than an ultra-long single trace, yet the paper only plots accuracy vs. tokens. A small wall-clock latency plot would make the practical win clear.

5) **Domain range:** The study is math-only; benchmarks on commonsense or coding tasks would show generality.


Question for my own understanding: Does the non-monotonic “overthinking” curve and parallel-thinking advantage still hold when you add a verifier?

---

> ### Author Rebuttal · Authors · 2025-07-31
>
> We thank the reviewer for the valuable feedback. We are happy that you found our work to have a **clear empirical observation**, an **insightful explanation** for the overthinking phenomenon, and a **simple, actionable fix** with our parallel thinking approach. We address the noted weaknesses below.
>
> > **Weakness 1:** Model scale: Results stop at 8 B params; a single run on a 30-ish B model (e.g., Qwen-30B) would help confirm that overthinking —and the parallel fix—persist at scale.
>
> **Response to Weakness 1.** Thank you for this point. We have conducted additional experiments using Qwen-3-32B thinking model and reported the results in Tables 1, 2 for GSM-8K and Tables 3, 4 for MATH-500 benchmarks.
>
> **Results on GSM-8K (Table 1 and Table 2 below):**
>
> **Table 1:** Wait & Think More Setup.
> | Average Thinking Tokens | Wait & Think Accuracy (%) |
> |------------------------|--------------------------|
> | 1182.4  | 93.25 (Standard Thinking) |
> | 1771.5  | 96.06 (↑) |
> | 2077.1  | 96.36 (↑) |
> | 2748.6  | 94.01 (↓) |
> | 4338.9  | 91.96 (↓) |
> | 5125.3  | 85.44 (↓) |
> | 7377.7  | 83.03 (↓) |
> | 9379.7  | 82.04 (↓) |
> | 11079.1 | 79.01 (↓) |
>
>
> **Table 2:** Parallel Thinking Setup.
>
> | Average Thinking Tokens | Parallel Thinking Accuracy (%) |
> |------------------------|--------------------------|
> | 1182.4   | 93.25 (Standard Thinking) |
> | 2064.3   | 93.71 (↑) |
> | 3506.9   | 94.69 (↑) |
> | 5561.4   | 95.45 (↑) |
> | 7908.6   | 96.54 (↑) |
> | 11388.5  | 97.42 (↑) |
> | 16593.1  | 97.42 (↔) |
>
>
> **Results on MATH-500 (Table 3 and Table 4 below):**
>
>
> **Table 3:** Wait & Think More Setup.
>
> | Average Thinking Tokens | Wait & Think Accuracy (%) |
> |------------------------|--------------------------|
> | 2897.2   | 92.2 (Standard Thinking) |
> | 4364.3   | 93.2 (↑) |
> | 6218.5   | 93.2 (↔) |
> | 7784.3   | 90.6 (↓) |
> | 9912.6   | 89.2 (↓) |
> | 12673.3  | 86.2 (↓) |
> | 14592.8  | 83.6 (↓) |
>
>
>
> **Table 4:** Parallel Thinking Setup.
>
> | Average Thinking Tokens | Parallel Thinking Accuracy (%) |
> |------------------------|--------------------------|
> | 2897.2   | 92.2 (Standard Thinking) |
> | 5441.9   | 93.2 (↑) |
> | 8230.1   | 93.6 (↑) |
> | 11348.8  | 93.6 (↔) |
> | 14085.6  | 94.2 (↑) |
>
> ***Takeaway:*** ***Our results persist at scale.*** Accuracy initially increases with extended thinking but subsequently declines. However, as evident from the results, with parallel thinking, we see a monotonic increase in accuracy.
>
> > **Weakness 2:** Predictive power missing: The variance story/ schematic discussed in Sec. 3 and Fig.4 is qualitative; it doesn’t give a formula to anticipate the overthinking peak across tasks or model scales. Without a way to estimate the tipping point in advance, practitioners still have to run experiments to find it.
>
> **Response to Weakness 2.** This is a great point. While a closed-form expression would be ideal, deriving one is extremely challenging in practice because a model’s tipping point depends on intricate interactions between architecture and task, which are hard to capture analytically without running inference. ***To address this challenge, we consider “parallel thinking”*** as a practical and robust strategy. This approach can be conceptualized as prioritizing the "breadth" of the search (exploring multiple independent reasoning paths) over its "depth" (a single, extended line of reasoning). By doing so, it effectively sidesteps the need to predict the tipping point. We also acknowledge that more complex, hybrid strategies could exist; for instance, future work could investigate optimal methods that combine both sequential extension and parallel sampling, creating a more sophisticated search algorithm.
>
> ***An entropy-driven approach to predict the tipping point:*** Our analysis in Fig. 5a–5d provides an empirical method: across GSM8K and MATH-500, we find that a sharp increase in the entropy of the model’s response distribution consistently precedes the performance drop. For example, entropy spikes after about $2^{11}$ tokens on GSM8K (Fig. 5a-5c) and after $2^{13}$ tokens on MATH-500 (Fig. 5b-5d). Monitoring response entropy can therefore serve as a useful proxy to anticipate the tipping point and halt reasoning early, offering a data-driven approach that mitigates overthinking without extensive experimentation or a theoretical formula.
>
> > **Weakness 3:** Decoding details: Entropy trends depend on sampling temperature/top-p; explicitly reporting those settings, or adding a robustness table, would strengthen Section 2.
>
> **Response to Weakness 3.** Thanks for this point. We used a sampling temperature of $0.6$ and set top-p as $0.95$.
>
> > **Weakness 4:** Latency & throughput: Because parallel branches can be batched, it is likely faster than an ultra-long single trace, yet the paper only plots accuracy vs. tokens. A small wall-clock latency plot would make the practical win clear.
>
> **Response to Weakness 4.** We agree with the reviewer and present the latency results in the form of a table to highlight the practical advantages of parallel thinking. For both "Wait & Think" and "Parallel" thinking, we used the same software and hardware configuration as mentioned in the Appendix (Section 2). We report the per-prompt average generation time for the GSM-8K benchmark using Deepseek-Distill-Qwen-1.5B.
>
> **Table 5:** We compare the average per-prompt generation time. As mentioned by the reviewer, we note that parallel thinking has less latency. Moreover, we remark that in the parallel thinking procedure, we can parallelize the sample generation to further reduce latency.
> |                | $2^8$ tokens | $2^{10}$ tokens | $2^{14}$ tokens |
> |----------------|--------------|---------------|---------------|
> | Wait & Think   | 29 secs      | 41 secs       | 173 secs      |
> | Parallel Thinking | 27 secs      | 35 secs       | 117 secs      |
>
> > **Weakness 5:** Domain range: The study is math-only; benchmarks on commonsense or coding tasks would show generality.
>
> **Response to Weakness 5.** We have added additional benchmarks, including coding (MBPP: Mostly Basic Python Problems Dataset) in Tables 6 and 7, and multi-task understanding (MMLU-Pro) in Tables 8 and 9, which further validates our hypothesis in the paper.
>
>
> ***Results on Multi-task Reasoning Task (Table 6 and Table 7):*** We report results for MMLU-Pro benchmark (TIGER-Lab/MMLU-Pro) on DeepSeek-R1-Qwen 1.5B using different setups:
>
> **Table 6:** Wait & Think More Setup.
>
> | Average Thinking Tokens | Wait & Think Accuracy (%) |
> |------------------------|--------------------------|
> | 637.2   | 30.00 (Standard Thinking) |
> | 980.0   | 34.28 (↑) |
> | 1071.6  | 38.57 (↑) |
> | 1702.9  | 35.71 (↑) |
> | 2005.7  | 35.71 (↔) |
> | 2144.5  | 44.28 (↑) |
> | 2629.2  | 41.42 (↓) |
> | 2944.9  | 40.00 (↓) |
> | 4874.4  | 34.28 (↓) |
> | 5120.5  | 30.00 (↓) |
>
>
> **Table 7:** Parallel Thinking Setup.
>
> | Average Thinking Tokens | Parallel Thinking Accuracy (%) |
> |------------------------|-------------------------------|
> | 637.2 | 30.00 (Standard Thinking) |
> | 1153.8 | 34.28 (↑) |
> | 2590.3 | 36.48 (↑) |
> | 3980.1 | 36.72 (↑) |
> | 5296.1 | 42.21 (↔) |
> | 9775.5 | 48.24 (↑) |
>
>
> ***Results on Code Generation Task (Table 8 and Table 9):***  We report results for the MBPP (Muennighoff/mbpp) Python code generation benchmark on Nvidia Llama-3.1-Nemotron-Nano-4B-v1.1 thinking model using different setups:
>
> **Table 8:** Wait & Think More Setup.
>
> | Average Thinking Tokens | Wait & Think Accuracy (%) |
> |------------------------|-------------------------------|
> | 3804.2 | 80.08 (Standard Thinking) |
> | 6435.1 | 78.02 (↓) |
> | 8889.4 | 78.02 (↔) |
> | 9484.3 | 74.94 (↓) |
> | 12306 | 70.42 (↓) |
>
> **Table 9:** Parallel Thinking Setup.
>
> | Average Thinking Tokens | Parallel Thinking Accuracy (%) |
> |------------------------|-------------------------------|
> | 3804.2 | 80.08 (Standard Thinking) |
> | 7262.4 | 80.08 (↔) |
> | 10941.2 | 80.08 (↔) |
> | 15862.8 | 80.18 (↑) |
> | 19010.3 | 80.18 (↔) |
>
> **Takeaway:** We observe that the "overthinking" phenomenon persists across these new domains, though its manifestation varies. On **MMLU-Pro**, our results show the familiar trend where accuracy **initially increases with extended thinking before declining**. However, on the **MBPP coding benchmark**, extended "Wait & Think" leads to a **monotonic performance decrease from the start**. Crucially, for both benchmarks, **parallel thinking proves to be a more robust strategy,** yielding stable or monotonically increasing accuracy and avoiding the performance drop associated with overthinking.
>
> >**Question:** Question for my own understanding: Does the non-monotonic “overthinking” curve and parallel-thinking advantage still hold when you add a verifier?
>
> **Response to Question.** Interesting point! In our work, we focused primarily on the setting of inference-time scaling without access to a true verifier, following the standard setup of verifier-free scenarios. In the presence of a true verifier, one can indeed leverage optimal decoding strategies such as controlled decoding or Best-of-N (BON) with verification, which we hypothesize can eliminate the overthinking curve.

---

> > ### Author Response · Authors · 2025-08-04
> > **Request to Reviewer 3KEc**
> >
> > Dear Reviewer 3KEc,
> >
> > We sincerely thank you again for your valuable feedback, which has greatly helped us improve the quality of our work. As the discussion deadline is approaching, we kindly request you to review our rebuttal.
> >
> > We are happy to address any remaining concerns. If our responses adequately address your concerns, we kindly ask you to consider adjusting the corresponding scores. Thank you again for your time and effort!
> >
> > Regards,
> >
> > Authors

---

> > > ### Comment · Reviewer_3KEc · 2025-08-06
> > > **Reviewer Response**
> > >
> > > Thank you authors for the rebuttal. Authors have addressed my concerns.
> > >
> > > Therefore, I increase my score to 4.
> > >
> > > Thank you.

---

### Comment · Area_Chair_xaNj · 2025-08-05
**[From AC] Reviewer Discussion Reminder**

Dear Reviewers,

Thank you for your time and effort in reviewing the paper.

As the reviewer-author discussion period ends on August 6 at 11:59 PM AoE, please take a moment to acknowledge the rebuttal and engage in the discussion if you haven’t.

Thank you again for your contributions to the review process.

Best,\
Area Chair

---

### Note · Authors · 2025-08-12

We are pleased that all reviewers appreciated the contributions, empirical findings, and systematic analysis in our work. We are also encouraged that reviewers find our paper **well-motivated** (Reviewer WH7z), **clearly written** (Reviewer f39d) with **well-conducted and systematic analysis** (Reviewer WH7z, f39d) and **clear empirical findings** (Reviewer 3KEc, f39d, dgbk). We would also like to thank all the reviewers for acknowledging our rebuttal and for increasing their scores towards acceptance.

| Reviewer | Post-rebuttal Comments                                                                                           |
|----------|----------------------------------------------------------------------------------------------------------|
| f39d     |  Reviewer acknowledged the rebuttal, and increased score to 5.
| dgbk     | Reviewer acknowledged the rebuttal, and increased score to 5. |
| 3KEc     | Reviewer acknowledged the rebuttal, and increased score to 4.                                                               |
| WH7z     | Reviewer acknowledged the rebuttal, and increased score towards acceptance.                                                 |



**Summary of discussion, new experiments, and updates.** We have addressed the reviewer's comments in individual responses to each reviewer. We are committed to incorporate the new experiments and all suggested changes in the final version of the manuscript. We summarize our discussions below.

+ [*Reviewer 3KEc, WH7z, f39d*] We extend our evaluations to large-scale models, specifically the Qwen-3-32B Thinking model, on GSM-8K and MATH-500. The results show that our findings persist at scale.
+ [*Reviewer 3KEc, WH7z*] Additional analysis on coding (MBPP: Mostly Basic Python Problems Dataset) and multi-task understanding (MMLU-Pro) benchmarks.
+ [*Reviewer 3KEc*] We added latency table to highlight the practical advantages of parallel thinking.
+ [*Reviewer WH7z*] We did additional experiments on higher-difficulty math benchmarks such as GPQA Diamond and AMC 2023.
+ [*Reviewer f39d*] We performed ablation studies on the effect of added prompts on overthinking behavior.
+ [*Reviewer dgbk*] A deeper understanding of overthinking behavior from an entropy perspective.
+ [*Reviewer dgbk*] We did additional analysis on the correlation between problem difficulty and the overthinking curve.

---

### Decision · Program_Chairs · 2025-09-17

**Decision:**

Accept (poster)

**Comment:**

This paper examines whether extending reasoning traces at test time consistently improves performance and finds a non-monotonic trend: accuracy initially rises but then declines due to overthinking, attributed to variance growth in the output distribution. To mitigate this, the authors propose parallel thinking, which distributes the token budget across multiple shorter reasoning paths and aggregates outputs, yielding consistent improvements across benchmarks and model scales.

The work is valuable and interesting for its clear empirical evidence and intuitive explanation. Some concerns remain, as the method overlaps with prior self-consistency approaches and the theoretical analysis is qualitative, but the rebuttal provided large-scale experiments, entropy-based predictors, prompt ablations, and broader benchmarks that addressed reviewer feedback and strengthened the contribution.

Thus I recommend acceptance.